

# Damping Identification of Offshore Wind Turbines using Operational Modal Analysis: A Review

Mees van Vondelen[1], Sachin T. Navalkar[2], Alexandros Iliopoulos[2], Daan van der Hoek[1], and Jan-Willem van Wingerden[1]

[1]Delft Center for Systems and Control, Delft University of Technology, 2628CN Delft, the Netherlands
[2]Siemens Gamesa Renewable Energy, Prinses Beatrixlaan 800, 2595BN Den Haag, the Netherlands

**Correspondence:** Sachin T. Navalkar (sachin.navalkar@siemensgamesa.com)

**Abstract.** To increase the contribution of offshore wind energy to the global energy mix in an economically sustainable manner, it is required to reduce the costs associated with the production and operation of offshore wind turbines (OWTs). One of the largest uncertainties and sources of design conservatism for OWTs is the determination of the global damping level of the OWT. Estimation of OWT damping based on field measurement data has hence been subject to considerable research

attention and is based on the use of (preferably operational) vibration data obtained from sensors mounted on the structure. As such, it is an output-only problem and can be addressed using state-of-the-art Operational Modal Analysis (OMA) techniques, reviewed in this paper. The evolution of classical time- and frequency-domain OMA techniques has been reviewed; however, literature shows that the OWT vibration data is often contaminated by rotor speed harmonics of significantly high energy located close to structural modes, which impede classical damping identification. Recent advances in OMA algorithms for

known or unknown harmonic frequencies can be used to improve identification in such cases. Further, the transmissibility family of OMA algorithms is purported to be insensitive to harmonics. Based on this review, a classification of OMA algorithms is made according to a set of novel suitability criteria, such that the OMA technique appropriate to the specific OWT vibration measurement setup may be selected. Finally, based on this literature review, it has been identified that the most attractive future path for OWT damping estimation lies in the combination of uncertain non-stationary harmonic frequency measurements with

statistical harmonic isolation to enhance classical OMA techniques, orthogonal removal of harmonics from measured vibration signals, and in the robustification of transmissibility-based techniques.

## 1 Introduction

The European Union has set a goal to reduce greenhouse gas emissions by 80-95% by 2050, (European Commission, 2011). Thus, nearly two-thirds of the energy production by 2050 is required to be supplied by renewable sources, of which (offshore)

wind is expected to be a major contributor. Globally, 2019 was a record year for offshore wind energy installation, with nearly 6.1 GW installed offshore and a cumulative global wind capacity of 29.1 GW. According to GWEC (2020), the total capacity will rise to 234 GW by 2030. Technical innovation is predicted to be one of the major drivers for this growth.



While the definition of external costs includes uncertainties, it has been shown by Alberici et al. (2014) that wind energy has the lowest external costs of all evaluated energy sources. The levelised cost of onshore wind energy is now at grid parity with the levelised cost of coal and natural gas, and in Europe lies within the range of 4.5-8.7 Eurocents/kWh. However, the levelised cost of offshore wind energy is between 6-11.1 Eurocents/kWh, which is more expensive than natural gas. As offshore wind energy in Europe enters a subsidy-free phase, it becomes more important for the wind industry to reduce costs to remain economically sustainable.

According to Blanco (2009), nearly 80% of project costs for onshore wind projects are directly related to the turbine costs. The tower and major support structure components can contribute to nearly 25% of the total turbine costs. For offshore wind projects, the contribution of the turbine costs is typically lower, roughly around 33% of the total project costs. However, offshore turbine foundations are correspondingly much more complex, contributing to nearly 21% of the total project costs. From these numbers, it is clear that the economic construction of turbine towers, foundations, and major support structure components can play a major role in determining the economic feasibility of an offshore wind energy project. Excessive conservatism in structural design is hence desired to be held to a minimum. Further, for offshore wind energy projects, operation and maintenance, including turbine downtime can contribute to up to 30% of the total project costs. Precise planning of such activities, and an ability to accurately predict the lifetime of turbine components, hence also forms an important factor influencing the economics of wind energy.

The design of wind turbine support structures is a multidisciplinary task that requires the evaluation of various loads faced by the turbine in its lifetime, including but not limited to wind loads, including extreme wind events, wave loads, including extreme sea states, seismic events, and loads exerted by self-weight and flexible deflections. As described in Nicholson et al. (2013), support structure design is then performed by considering stress analyses, including stress concentrations, buckling analyses, extreme deflection analysis, soil bearing capacity calculations, and balancing these analyses against practical considerations such as manufacturability, transportability, and resonance avoidance considerations. According to Van der Tempel (2006), support structure analysis can be performed in the frequency domain, but the calculation of structural loads as an initial step may also be performed in the time domain, using dedicated aeroelastic simulation environments.

As per Rao (2005), structures like wind turbines demonstrate several 'modes' of natural vibration, each associated with a natural frequency of oscillation. If the structure receives input energy, such as from wind or wave loading, at a frequency close to its natural frequency of oscillation, 'resonant' behaviour is said to occur, and significant amplitudes of oscillation are observed. Sustained high-amplitude oscillations, may cause progressive failure or 'fatigue' of the structure. Design for a sufficient fatigue lifetime (typically 20-30 years) is one of the key requirements for turbine structural design.

One of the most important structural properties that is required to be defined in both fatigue and extreme loads analysis of wind turbines is the damping ratio of the structure. The damping ratio or damping of the structure is proportional to the rate of decrease of the frequency response function when close to the natural frequency, (Rao, 2005). Default values are often assumed in finite-element or multi-body simulations for damping ratios of vibrating structures. However, the fatigue life of vibrating structures shows a high sensitivity to the assumed damping ratio, and the structural damping needs to be chosen with care, for instance using experimental data as shown in Kihm et al. (2018).





**Table 1.** Prescribed structural damping by different standards (Van der Tempel, 2006).

| Standard | Damping (% critical) |
|---|---|
| ISO 19902 DIS (2004) | For fatigue: 1-2% |
| American Petroleum Institute (2000) | For fatigue: 2% |
| Germanischer Lloyd (2000) | 1% |
| DNV-OS-J101 (2004) | 1% |

All phenomena that add damping to the Offshore Wind Turbine (OWT) structure have been enumerated in Koukoura et al. (2015):

1. Aerodynamic damping, or damping introduced by the aeroelastic interaction of the incident wind on the turbine structure.

2. Control damping, or damping introduced by the controller actively in feedback to various measured quantities such as rotor speed, nacelle acceleration, or blade loads. The damping is effectuated through actuators such as generator torque, active blade pitch, or other more advanced actuators.

3. Mass or liquid dampers. Such dampers are often specifically designed to add damping to specific modes of the turbine
structure and are typically located near the tower top, where motions are maximal.

4. Structural damping, or material damping. This damping is an inherent property of any physical structure.

5. Hydrodynamic damping, or radiation damping. For offshore turbines, the motion of the submerged components of the support structure is directly connected with this kind of damping.

6. Soil damping, or damping induced by the soil on the piles driven into the sea bed.

Of all sources of damping, items 1-3 are often implicitly included in most modern turbine simulation environments that work in the time domain. When working in the frequency domain, it often becomes necessary to include the effect as an explicit additional damping ratio, see for instance the work of Van der Tempel (2006) on the inclusion of aerodynamic damping in frequency-domain support structure models. Structural damping has to be included explicitly as an additional material damping ratio in most simulation environments. Hydrodynamic and soil damping may be included implicitly in the form of additional
dashpots in the structural model, or they may be subsumed directly into an explicit structural damping ratio. Please refer to Versteijlen et al. (2011) for an in-depth treatment of the modelling of soil damping.

Depending on the damping components included, the explicit structural damping ratio used for structural design is defined by various standards and codes to lie within 1-5% critical damping, with the lower values typically recommended for fatigue design, Van der Tempel (2006). The paper summarises the damping recommendations of various standards as per Table 1.
Although these damping values are prescribed to be used for design, they may be considerably more conservative than the actual damping found in the field. For instance, field experiments from Versteijlen et al. (2011) specifically show that the





soil damping found experimentally is typically higher than that used by the industry. Further, Martinez-Luengo et al. (2016) mentions that overall turbine damping may undergo changes in turbine lifetime. For more accurate and economical design, and for the precise monitoring of fleet turbine structural lifetimes, it is hence necessary to estimate turbine damping directly from measured data. It should be noted here that all forms of OWT damping are considered in the rest of the paper to be subsumed under the term 'structural damping'. The identification of individual forms of damping using measurement data is not covered here.

The identification of damping from measurement data can be performed following two philosophies: Experimental Modal Analysis (EMA) and Operational Modal Analysis (OMA). EMA uses input and output data from dedicated system experiments, to obtain estimates of system damping. Such experiments may involve the use of dedicated vibration shakers or impact hammers such that the input excitation or impulse is known. A recent example of the use of EMA for determining ice mass accumulation on wind turbine blades has been given in Gantasala et al. (2018), where both vibration shakers and impact hammers are used. With sufficient instrumentation, the data collected during such an experiment is rich in information, and it is possible to estimate modal frequencies and damping ratios, and even the mode shapes themselves. One possibility to extend this methodology for OWT damping estimation is the use of transient load cases such as a rotor stop test Devriendt et al. (2013) to generate an OWT impulse reponse.

However, generating impulse responses only permits OWT damping estimation in the idling state. In general, it is expensive or infeasible to artificially excite the OWT in normal operation or to measure all sources of input excitation. Hence, the estimation philosophy of OMA is considered in literature to be a more viable alternative for damping estimation (Devriendt et al., 2013). As the name suggests, this approach uses turbine data from normal operation for damping estimation. This approach necessarily requires the use of output-only data and makes the assumption that all input sources of excitation are Gaussian white noise and therefore by definition wide-sense stationary with zero mean and finite variance.

While the scope of this paper is limited to a review of OMA algorithms in the context of damping estimation of offshore wind turbines, it needs to be noted that the inspiration for several classical OMA algorithms originates from EMA techniques. EMA techniques could be directly adapted for OMA approaches using the Natural Excitation Technique (NExT) developed by James et al. (1995), which is able to convert stationary output signals from the plant into decaying impulse response signals, directly amenable to a log-decrement analysis of system damping. Another breakthrough for systems with multiple modes, that are still able to admit a Linear Time-Invariant (LTI) system realisation, came with the development of the Stochastic Subspace Identification (SSI) family of algorithms (Van Overschee and De Moor, 1991). These algorithms provide a noniterative solution to the identification problem, using numerically stable linear algebra techniques like the QR and singular value decompositions.

However, significant challenges persist in the use of OMA for the damping identification of offshore wind turbines. One of the main assumptions of most OMA algorithms is that the plant is LTI. However, as detailed in Ozbek et al. (2013), due to several dynamic characteristics of the wind turbine, including rotor rotation and wind speed-based variation of plant dynamics, this OMA requirement is not strictly fulfilled by offshore wind turbines. Further, due to the presence of rotor speed-dependent harmonics in measured data, the white-noise assumption of input excitation is violated.



If data is used only from an idling OWT, the issue of harmonic interference does not occur, (Devriendt et al., 2014; Kramers et al., 2016; Van der Hoek, 2017). Here, the LTI approximation is relatively good, and the input excitation is close to random. However, using only idling data severely restricts the amount of data available. Further, a change in soil or structural dynamics that may occur under operation cannot be estimated. Hence, operational data for damping estimation is often preferred, especially for structural health monitoring purposes where structural issues may be more visible under operation.

For stationary harmonics, many OMA algorithms directly identify the harmonic peaks as the ones with zero damping. However, as shown in El-Kafafy et al. (2014), as a result of temporal variations in turbine rotor speed, the harmonic peaks seen in a finite set of data may often appear damped and difficult to distinguish from structural peaks. A second issue occurs when the frequency of the harmonic coincides with structural frequencies. Classical OMA algorithms may fail to differentiate between the two, and the structural mode may be estimated based on either mode or an incorrect combination of the two. Finally, if the harmonic energy is significantly high, classical OMA algorithms may fail entirely in identifying a structural mode close to the harmonic. In recent literature, researchers have developed several methods to circumvent these problems, or include harmonics explicitly in the upgraded estimation algorithms. This paper provides an overview and comparison of OMA algorithms intended to handle the challenges raised in the damping identification of offshore wind turbines.

The rest of the paper is organised as follows: Section 2 gives an overview of the wind turbine structure, and details the instrumentation for data collection. Section 3 discusses classical OMA algorithms, developed for general model parameter estimation. Section 4 specifically discusses the case where the harmonics are known and can be directly integrated into OMA algorithms. Section 5 does not assume known harmonics, but instead defines statistical and other pre-processing techniques that can be used to enhance damping identification. Section 6 describes transmissibility-based identification algorithms which should in principle be entirely insensitive to harmonic interference. Section 7 defines suitability criteria that may aid in the choice of a suitable OMA algorithm for the specific use cases. Finally, Section 8 provides a practical implementation discussion and Section 9 concludes this review paper.

## 2 Context and Definition of the Damping Identification Problem

The OWT is a complex assembly of several components that demonstrate several loads-relevant modes of vibration. This section briefly summarises the most relevant OWT structural components and the relevant modes for the damping identification problem. Further, the instrumentation desired and necessary for high fidelity estimation of OWT structural damping is also enumerated.

### 2.1 Components of an Offshore Wind Turbine

For support structure design, an OWT may be considered to consist of the following main components as shown in Fig. 1 (Van der Tempel, 2006):

- Rotor: This is the rotating part of the wind turbine, typically consisting of three rotor blades mounted on a hub. The blade may be pitched along its longitudinal axis using hydraulics or electrical motors.





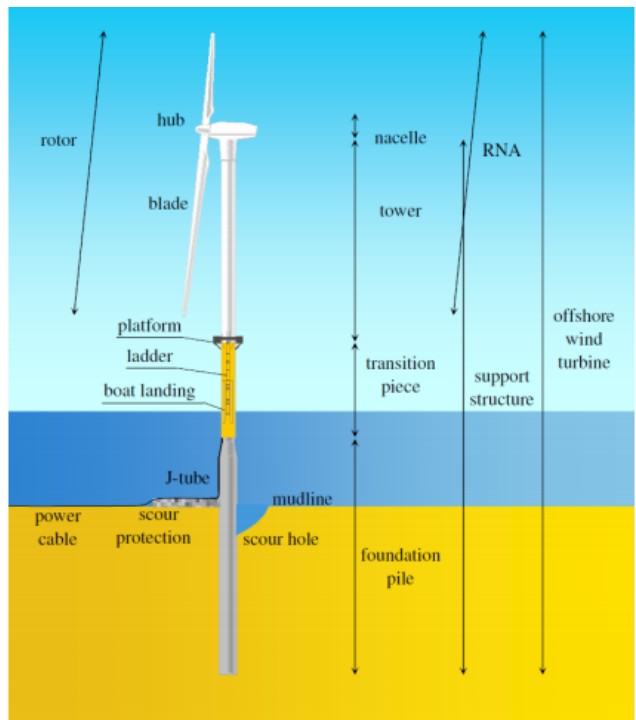

**Figure 1.** Components of an OWT relevant for support structure design (Van der Tempel, 2006).

- Nacelle: This is the housing of the shaft and the drive train, including the generator and gearbox, if any. The nacelle is able to yaw in such a manner that the rotor can be oriented as desired with respect to the wind direction. Together, the

Rotor and Nacelle form the Rotor Nacelle Assembly (RNA) of the wind turbine.

- Tower: Onshore, this is the part of the turbine support structure located above the ground. Offshore, this part of the support structure is typically located above sea level and connects via a transition piece to the offshore foundation.

- Foundation: The foundation is the part of the support structure embedded directly into the seabed. Foundations may be of multiple types as shown in Fig. 2, from left to right a monopile foundation, a gravity-based foundation, a tripod, and a

jacket-type foundation. By far the most commonly installed foundation type for offshore wind projects is the monopile foundation. However, as water depths increase for offshore wind projects, it is expected that more complex foundation types will become more widely used. The here depicted foundation types are all of the fixed-bottom category. Floating foundations are not covered within the scope of the current review paper.

The integrated wind turbine structure comprising the above-mentioned components shows different loads-relevant dynamic

structural modes described in the next subsection.





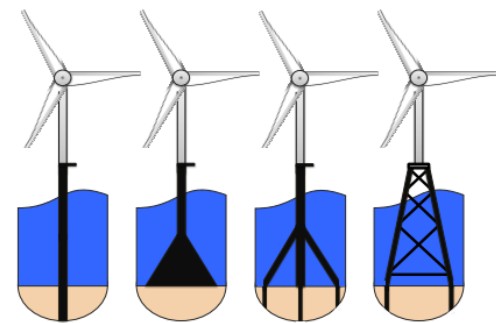

**Figure 2.** From left to right, a monopile foundation, a gravity-based foundation, a tripod, and a jacket-type foundation (Van der Valk, 2014)
.

**Figure 3.** Dominant modes of an OWT. The fore-aft and side-side bending pairs are significantly close to each other (El-Kafafy et al., 2014)
.

## 2.2 Modal behaviour and harmonics of an Offshore Wind Turbine

As the objective of this paper is to identify the modal damping of an OWT from measured data, it is first required to define the turbine modes that are most relevant for the loads derived from turbine aeroelastic simulations. As described in El-Kafafy et al. (2014), the first five structural modes of an OWT have been depicted in Fig. 3. These modes are listed below:

  – $1^{\text{st}}$ Fore-Aft and Side-Side bending modes.

  – Coupled blade-tower mode.

  – $2^{\text{nd}}$ Fore-Aft and Side-Side bending modes.

Apart from these modes, the first two torsion modes typically fall within the frequency range of interest. For an OWT design with a monopile foundation, torsion is rarely design-driving and torsional damping has not been studied to a significant extent
in literature. However, precise modelling of torsion may become relevant for more complex foundation types.



The third fore-aft and side-side bending modes are typically seen to reveal higher damping and correspondingly lower contribution to the overall structural vibrational energy. It is expected that the algorithms reviewed in this paper can be extended directly to the identification of third bending mode damping, however, these modes have also not received significant attention in current literature. With larger and more flexible OWT structures, it is expected that the contribution of these modes to total fatigue lifetime will increase. Other structural modes, especially those corresponding to frequencies higher than 5 Hz, are expected to have a low impact on the overall turbine dynamic behaviour and have hence not been studied to a significant extent in the context of OWT global damping estimation.

From Fig. 3 it can be observed that some of the structural modes are relatively close to each other. This modal proximity may cause issues in damping estimation. As described by Ozbek et al. (2013), the estimation problem is also complicated by the following harmonic interference (here 'P' refers to the fundamental period of rotor rotation):



– 1P: This harmonic originates from mass, pitch, and other rotor imbalances. This harmonic may interfere with the first turbine bending modes, especially when the tower is of the so-called 'soft' type (first eigenfrequency lower than 1P).

– 3P: This harmonic originates from the rotational sampling of turbulence, wind shear, tower shadow, and other aerodynamic phenomena. This harmonic typically interferes with the first turbine bending modes for low wind speeds.


– 6P: This is a multiple of the 3P harmonic, and may interfere with the second turbine bending modes in practice.

– Higher harmonics are usually not relevant for damping identification, although 9P, 12P, and so on may interfere with the third structural bending modes of the OWT.

The above-defined structural modes and rotor harmonics can be observed to varying extents in the measured data available from the OWT, as described in the next subsection.


## 2.3 Measurement data acquisition

The structural modes described in the previous section are typically captured by the tower top/nacelle accelerometers that are mandated by IEC61400 to be placed in all commercial OWTs. As such, the information regarding structural frequencies and damping may be possible to be extracted from OWT SCADA data as collected by most turbine manufacturers and wind farm owners. However, these IEC-mandated sensors may not be adequate for the current purpose for the following reasons:


– Sampling frequency: To extract the requisite information from such SCADA data, it is required that the sampling frequency of the data must be at least 5-10 Hz. In order to save storage space, it is common practice to decimate the data to ten-minute signal statistics, which is clearly inadequate to perform dynamic analyses.

– Low-frequency fidelity: Accelerometers may have a high-pass behaviour that may affect the frequency range of interest. It is required that at least the frequency range between 0.1-5 Hz is accurately represented in the measured data.


– Signal-to-Noise Ratio (SNR): While these accelerometers typically have sufficient range to capture high acceleration levels, they may not for economical reasons have adequate resolution to be able to capture low acceleration levels.





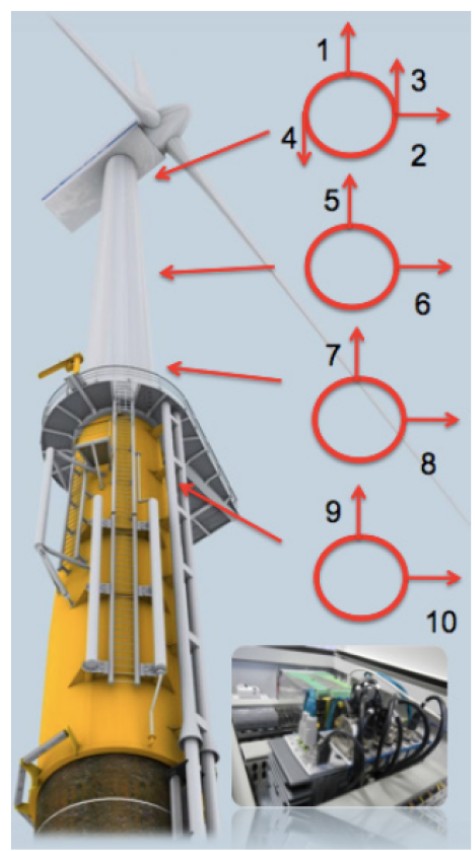

**Figure 4.** Example of tower instrumentation for damping estimation, with sensor locations and orientations (Manzato et al., 2014).

Idling at low wind speeds leads to relatively low levels of tower top motion, and the SNR of commercial tower top accelerometers may be far too low to extract useful modal information from this data.

– Modal amplitude: While the first bending modes of an OWT show maximal amplitude at the tower top, this is not typically the case for the higher bending modes. As such, a location midway down the tower may show cleaner signals with lower SNR that may be more suitable for data collection than the default tower top accelerometers.

While additional sensors add significant cost, dedicated damping measurement campaigns like Devriendt et al. (2013); Manzato et al. (2014) often use a much more heavily instrumented tower. Figure 4 shows a tower that is equipped with 10 accelerometers, divided over 4 levels, with each level measuring horizontal plane accelerations, and the top level also measuring torsion. It should be noted that as the nacelle is permitted to yaw during the campaign, the measured accelerometer signals shall no longer be strictly fore-aft or side-side, but necessitate a coordinate transformation in order to retrieve the fore-aft and side-side behaviour of the tower. As such, it is also necessary for the yaw angle measurement to be available for this coordinate transformation, and that this measurement is correctly synchronised with the accelerometer data.



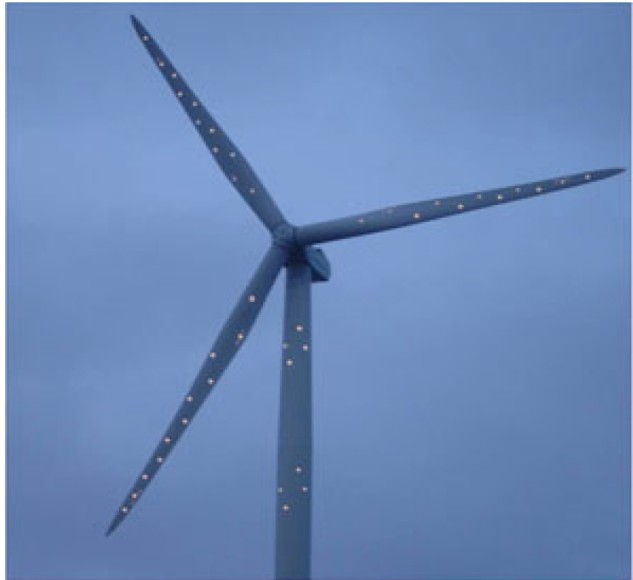

**Figure 5.** Example of turbine instrumentation with reflective markers for photogrammetry or laser interferometry (Ozbek and Rixen, 2013).

.

Other sensors that can be used for structural health monitoring and damping estimation have been reviewed by Wymore et
al. (2015). Apart from accelerometers, the tower can also be instrumented with displacement sensors or strain sensors. Strain
sensors show the highest sensitivity and SNR when located at the base of the tower or the root of the blades, and these may be
of the electrical or optical (Fibre-Bragg) type, which show different reliability and temperature sensitivity characteristics.

Locating sensors on the foundation has so far been limited to research applications since durability and accessibility can be
a major issue. These sensors may be of the same type as tower sensors, (i.e. strain or acceleration sensors), but they require
special packaging to withstand being submerged in seawater, and they may require diver-based maintenance.

Nontraditional sensors have also been reviewed in Wymore et al. (2015); for instance, a microwave radar-based system has
been reviewed that is able to detect modal behaviour from a distance of 1 km. Lidar solutions have also been proposed for data
collection for modal analysis. Ozbek and Rixen (2013) demonstrate the use of reflective markers located on a 2.5 MW turbine
structure for the purpose of modal analysis, as shown in Fig. 5. These markers do not require any further electrical or optical
instrumentation on the turbine, but can be used for deformation measurement using photogrammetric principles. With the use
of cameras located roughly 220 m away from the turbine, it is possible to measure marker deformation with a resolution of 2.5
mm. At standstill, this method shows low SNR, but it can be replaced by laser interferometry measurements. In this technique,
a laser beam is projected on different markers, and the reflected light is collected. Based on the change in the frequency of the
reflected beam, the velocity of the marker can be determined. Although these techniques have been used in the paper for an
onshore turbine, an offshore application has so far not been reported.




While input excitations, in the form of wind and wave measurements, can also be incorporated in the estimation of structural damping, the remaining part of the paper focusses on output-only damping estimation and hence a description of such sensors is omitted.

## 3    Classical Operational Modal Analysis Algorithms

Turbine vibration data available via measurement setups as described in the previous section can be used to perform modal analysis and damping estimation of the turbine structure. The dynamics of an OWT can be modelled using first principles, and the unknown model parameters, including structural damping, can be estimated from measurement data using system identification techniques described in Ljung (1987). However, the damping term often appears nonlinearly in such first-principles models of structural systems, and requires an iterative estimation method. Furthermore, unless good initial estimates are avail-

able, the optimisation of such nonlinear cost functions can get stuck in local minima that give incorrect damping estimates. The identification of damping from vibration data is also an application where the input to the dynamic system (i.e. the OWT) is typically unknown or unmeasurable. For this reason, the focus of the current and subsequent sections is on identification techniques that can handle output-only identification, and preferably use linear least squares formulations or noniterative linear algebra techniques to arrive at the damping estimate of the OWT structure.

'Classical' OMA algorithms are covered in this section, and they are defined as algorithms that do not pay specific attention to the presence of rotor harmonics in measurement data. Chronologically, these precede and form the basis for the algorithms that are described in subsequent sections which are able to specifically address rotor harmonics. Classical OMA may perform damping estimation either from time-domain or frequency-domain data, Rainieri and Fabbrocino (2014).

In the rest of the paper, the term 'poles' of the identified system and 'eigenvalues' of the state evolution matrix correspond to

physical or fictional modes of the dynamic system identified using measured data. The natural frequency of the mode is given by the magnitude of the pole or eigenvalue, while the ratio between the real part of the pole and the natural frequency provides a measure of modal damping.

### 3.1    Time-domain algorithms

As also described in the introduction, the first time-domain algorithms described in literature perform EMA, which requires

the structure to be excited using a vibration shaker or an impact hammer. The impulse response and the free decay or natural vibrations are used for damping estimation in Ibrahim (1973), in the Ibrahim Time Domain (ITD) method where the damping of different modes is identified based on multiple impulse response measurement stations on a vibrating structure. Zaghlool (1980) extends this work by showing that with an accelerometer at a single measurement station on a vibrating structure, it is possible to deduce all relevant modal frequencies and dampings using impulse response data, as long as all such modes are

observable at that station. This approach, named the Single-Station Time-Domain (SSTD) technique, assembles the discretely





sampled available acceleration data $\ddot{y}_k$ in the following manner:

$$\ddot{Y}(t_1) = \begin{bmatrix} \ddot{y}_{t_1} & \ddot{y}_{t_2} & \cdots & \ddot{y}_{t_{2n}} \\ \ddot{y}_{t_1+T} & \ddot{y}_{t_2+T} & \cdots & \ddot{y}_{t_{2n}+T} \\ \vdots & \vdots & \ddots & \vdots \\ \ddot{y}_{t_1+T} & \ddot{y}_{t_2+T} & \cdots & \ddot{y}_{t_{2n}+T} \end{bmatrix}, \ \ddot{\dot{Y}}(t_1) = \ddot{Y}(t+\Delta t), \ \ddot{\ddot{Y}}(t_1) = \ddot{\dot{Y}}(t+\Delta t). \tag{1}$$

Here, $T$ and $\Delta t$ are arbitrary time lags. The modal frequencies and dampings of the dynamic system can be estimated as the eigenvalues of the following matrix:


$$Z = \begin{bmatrix} \ddot{\dot{Y}} \\ \ddot{Y} \end{bmatrix} \begin{bmatrix} \ddot{\ddot{Y}} \\ \ddot{\dot{Y}} \end{bmatrix}^{-1}. \tag{2}$$

While these expressions refer to acceleration data, they can also be used for velocity or displacement response data.

Brown et al. (1979) worked specifically on Single-Input Single-Output (SISO) systems, or on multivariable systems that can be decomposed into decoupled SISO systems. This paper lays out the framework of the Complex Exponential algorithm which uses the discretely sampled experimental impulse response $y_k$ to deduce a set of autoregressive coefficients $a_0 \cdots a_{2n-1}$ for the

underlying dynamic system representation of the structure, in the following manner:

$$\begin{bmatrix} y_{2n-1} & y_{2n} & \cdots & y_{4n-2} \\ y_{2n-2} & y_{2n-1} & \cdots & y_{4n-3} \\ \vdots & \vdots & \ddots & \vdots \\ y_0 & y_1 & \cdots & y_{2n-1} \end{bmatrix} \begin{bmatrix} a_0 \\ a_1 \\ \vdots \\ a_{2n-1} \end{bmatrix} = \begin{bmatrix} y_{4n-1} \\ y_{4n-2} \\ \vdots \\ y_{2n} \end{bmatrix}. \tag{3}$$

With the autoregressive coefficients, the eigenvalues can be determined as the complex roots of:

$$\sum_{k=0}^{2n} a_k x^k = 0, \tag{4}$$

where $a_{2n} = 1$ without loss of generality. It follows per definition that the damping of each mode is the negative of the nor-

malised real part of the corresponding eigenvalue.

It is interesting to note that in this algorithm, as in many subsequent algorithms, the chosen order of the autoregressive system $n$ is a user choice, and the rank of the data matrix may be used as an indication of model order. Brown et al. (1979) recommend using one and a half times the number of expected modes as the value of $n$. Per definition, this approach will find artificial or mathematical poles as well as physical poles. Artificial poles are likely to be found in areas of low mode density,

and the algorithm may be unable to identify closely-spaced modes with fidelity. Filtering of data is suggested as a palliative for this issue, and indeed this may also improve the performance of several approaches mentioned in this paper. In this algorithm, using different data samples may lead to variance in the identified modes. Hence, a least-squares approach, the Least Squares Complex Exponential (LSCE) algorithm, has also been presented in this paper to improve the robustness of the solution.





The above algorithms deal specifically with one single output channel. For the case where multiple output channels are available from the measurement system on the vibrating structure, the Eigensystem Realisation Algorithm (ERA) was developed by Juang and Pappa (1985). In this algorithm, given the multivariable impulse response of the system $Y_k \in \mathbb{R}^{n_y}$, the following (block Hankel) matrix can be defined:

$$H_{k-1}(r,s) = \begin{bmatrix} Y_k & Y_{k+1} & \cdots & Y_{k+s-1} \\ Y_{k+1} & Y_{k+2} & \cdots & Y_{k+s} \\ \vdots & \vdots & \ddots & \vdots \\ Y_{k+r-1} & Y_{k+r} & \cdots & Y_{k+r+s-2} \end{bmatrix}. \tag{5}$$

The state evolution $A$ matrix of the underlying dynamic system can be estimated from a Singular Value Decomposition (SVD) of the data matrix in the following manner:

$$H_0(r,s) = U\Sigma V^T, \tag{6}$$

$$A = \Sigma^{-\frac{1}{2}} U^T H_1(r,s) V \Sigma^{-\frac{1}{2}}. \tag{7}$$

The eigenvalues of $A$ give the modal frequencies and damping of the system. Here, it is important to note that the pair $(r,s)$ needs to be larger than the expected order $n$ of the system. For an ideal system with no noise, the $(n+1)^{\text{th}}$, and higher singular values in the diagonal matrix $\Sigma$ will be zero. Practically, due to noise, these values will be nonzero. However, the order of the system $n$ can typically still be identified by a jump in the magnitude of the singular values of this matrix. Hence, the eigenvalues of $A$ beyond the first $n$ modes can be neglected, as they are assumed to be artificial. This order identification step is widely used in algorithms that rely on an SVD step in the identification procedure.

While in this review paper it is assumed that the OWT is LTI, this assumption can be relaxed if one were to use the extension of ERA for Linear Time-Varying (LTV) systems according to Majji et al. (2010). This algorithm uses a co-ordinate transformation to enable the use of ERA for LTV systems.

The literature above requires impulse response data to generate damping estimates. However, generating a pure OWT impulse response, uninfluenced by wind and waves, is not feasible. Hence, it is preferable to use operational turbine data for damping estimation. It was proven by James et al. (1995) that the cross-correlation $R_{ijk}(t)$ of two output sequences $i$ and $j$, given an unknown input $k$ shows the same frequencies and exponential decay as the impulse response of the structure. If the input $k$ is a zero-mean white-noise signal of constant covariance $\alpha_k$,

$$R_{ijk}(t) = \sum_{r=1}^{n} A_{ijk}^r e^{-\zeta^r \omega_N^r t} \cos\omega_D^r t + B_{ijk}^r e^{-\zeta^r \omega_N^r t} \sin\omega_D^r t, \tag{8}$$

$$\begin{bmatrix} A_{ijk}^r \\ B_{ijk}^r \end{bmatrix} = \sum_{s=1}^{n} \frac{\alpha_k \Psi_{ir} \Psi_{kr} \Psi_{js} \Psi_{ks}}{m^r \omega_D^r m^s \omega_D^s} \int_0^\infty e^{(-\zeta^r \omega_N^r - \zeta^s \omega_N^s)\lambda} \sin\omega_D^s \lambda \begin{bmatrix} \sin\omega_D^r \lambda \\ \cos\omega_D^r \lambda \end{bmatrix} d\lambda. \tag{9}$$

Here, $\zeta^r$, $\omega_N^r$, $\omega_D^r$ and $m^r$ are the damping, natural frequency, damped natural frequency, and modal mass corresponding to the $r^{\text{th}}$ mode. Further, $\Psi_*$ corresponds to the constant modal matrix that transforms structural model co-ordinates into the modal co-ordinates of the system.





Given this insight that cross-correlation of two output sequences behaves in the time domain in the same manner as an impulse response, James et al. (1995) extend the impulse response-based modal analysis methods like ERA, described above, with the Natural Excitation Technique (NExT), and can successfully perform the damping estimation of a wind turbine based
purely on operational output data. The NExT algorithm has also been combined with the LSCE algorithm and used for the identification of OWT damping in Ozbek and Rixen (2013) and Ozbek et al. (2013). The authors identify the first two modal frequencies and damping of an OWT from operational data, however, they point out that significant variance is seen in the fore-aft damping estimate. It must be noted here that the NExT technique requires at least two concurrent sets of measurement channels (e.g. two sets of strain sensors at different structural locations), of which all relevant structural modes must be
observable in at least one of these measurement channels.

As distinct from the methods described above, subspace identification does not treat the underlying system as an input-output transfer matrix but focusses on attaining a (minimal) state-space realisation of the plant. This identification technique relies on the principle that the state of the system can be retrieved from the measured data. Based on such a (Kalman-optimal) state estimate, the state evolution matrix $A$ of the system can be estimated. As they work with state-space models, all subspace
identification algorithms are inherently multivariable. For the MOESP (Multivariable Output Error State Space) and the N4SID (Numerical algorithms for Subspace State-Space System IDentification) families of subspace identification algorithms, it has been formally shown by Verhaegen, M. and Dewilde P. (1992) and Van Overschee and De Moor (1994) respectively that asymptotically unbiased estimates of model parameters are achieved as long as the system input has adequate persistency of excitation. These two concepts are relevant for all identification techniques reviewed herein:

– Damping estimation of an OWT is only possible if the (unmeasurable) inputs sufficiently excite all structural modes of interest. For a typical operating wind turbine, the wind/wave excitation is typically sufficient to excite the lower structural modes, but the excitation response at the third and higher structural modes may not contain enough energy to separate them from measurement noise.

– Asymptotic unbiasedness implies that a sufficient amount of data is necessary to perform the damping identification.
Although formal proofs do not exist for all methods reviewed, the quality of damping estimation improves with an increasing amount of measurement data for most algorithms. Ozbek et al. (2013) recommend that for the LSCE and SSI algorithms, a minimum of 200 cycles of the lowest frequency of interest are required for adequate identification quality, for the application of OWT damping estimation.

Verhaegen, M. and Dewilde P. (1992) also prove formally that the MOESP approach is less sensitive to measurement noise as
compared to the non-subspace methods, especially for lightly-damped structures.

Stochastic Subspace Identification (SSI) algorithms, as developed by Van Overschee and De Moor (1991) are suitable for the damping identification of structures subject to unmeasurable disturbances. SSI arranges the obtained measurement data





$Y_k \in \mathbb{R}^{n_y}$ into so-called 'past' and 'future' block Hankel data matrices, $H(0, i-1)$ and $H(i, 2i-1)$ defined as follows:

$$H(k,i) = \begin{bmatrix} Y_k & Y_{k+1} & \cdots & Y_{k+j-1} \\ Y_{k+1} & Y_{k+2} & \cdots & Y_{k+j} \\ \vdots & \vdots & \ddots & \vdots \\ Y_{k+i} & Y_{k+i+1} & \cdots & Y_{k+i+j-1} \end{bmatrix}. \tag{10}$$

For the construction of the 'past' Hankel data matrix $H(0, i-1)$, it is sufficient to consider a reduced number of reference outputs as long as observability is retained, Peeters and De Roeck (1999). The column size of both Hankel matrices remains $j$, and the quality of the damping estimate increases as $j \to \infty$. Further, as in ERA, the choice of $i$ must be made such that it is larger than the expected model order of the underlying system.

The primary postulate of the SSI algorithms is that the state can be retrieved from the row space of the orthogonal projection
of the row space of the 'future' data on the row space of the 'past' data, as long as the unknown input excitation is zero-mean white noise. Under the approximation that $j \to \infty$ is valid, the state of the system can be retrieved from the matrix $Z_i$ in the following:

$$E[H(i, 2i-1)H(0, i-1)^T](E[H(0, i-1)H(0, i-1)^T])^{-1}H(0, i-1) = \mathcal{O}_i \underbrace{\mathcal{C}_i L_i^{-1} H_{0,i-1}}_{Z_i}. \tag{11}$$

Here, $E[*]$ is the expectation operator for stochastic signals. Thus, the row space of the left-hand side of the above equation
gives the Kalman optimal state estimates of the system. In the equation above, $L_i \in \mathbb{R}^{in_y \times in_y}$ is the constant output covariance matrix and the (unknown) system forward and backward observability matrices are defined as:

$$\mathcal{O}_i^T = \begin{bmatrix} C^T & A^T C^T & \cdots & (A^{i-1})^T C^T \end{bmatrix}, \tag{12}$$

$$\mathcal{C}_i^T = \begin{bmatrix} A^{i-1}G & A^{i-2}G & \cdots & G \end{bmatrix}. \tag{13}$$

Here, $A \in \mathbb{R}^{n \times n}$ is the state evolution matrix to be determined, $C \in \mathbb{R}^{n_y \times n}$ is the unknown system output matrix, and $G \in$
$\mathbb{R}^{n \times n_y}$ is the unknown backwards Kalman gain. Once the states $Z_i$ have been estimated, $A$ is given by:

$$A = E[Z_{i+1} Z_i^T](E[Z_i Z_i^T])^{-1}. \tag{14}$$

Here it needs to be noted that neither the states $Z_i$ nor the matrix $A$ are unique. Subspace identification algorithms can only estimate the states and the state evolution matrix up to a similarity transformation. However, as the system eigenvalues remain invariant under similarity transformations, this limitation is not expected to affect the estimation of structural damping. The
covariance of the resultant SSI damping estimates has been defined by Reynders et al. (2008) and validated by Reynders et al. (2016).

One of the conditions for low estimate covariance is that the row size $i$ of the Hankel data matrix is required to be larger than the expected order $n$ of the system. However, excessively large model order estimates may lead to overfitting of the measurement data. Van der Veen et al. (2013) show that the Akaike Information Criterion can be used to deliver a good first
estimate of this SSI identification hyperparameter.





Ozbek et al. (2013) compare the performance of SSI and LSCE using OWT simulation data obtained using a nonlinear aeroelastic tool. In Kramers et al. (2016) and Van der Hoek (2017), MOESP and Predictor-Based Subspace Identification (PBSID) SSI approaches (Van der Veen et al., 2013) were used for damping estimation of OWTs from field data, respectively. In both cases, the measurement data was obtained from an idling wind turbine, hence no harmonics were present in the data
and first mode damping estimation could be performed successfully.

While this subsection focussed on the use of time-domain data for the estimation of system damping, frequency-domain data has also been classically used for this purpose, as summarised in the next subsection.

## 3.2  Frequency-domain algorithms

While time-domain algorithms extract damping information from time-domain data, the techniques introduced in this section
use the Frequency Response Function (FRF) or frequency-domain vibration data to perform modal analysis. While these techniques can make direct use of operational OWT data, they make the implicit assumption that the underlying dynamical system is LTI. For an OWT, this implies that the data collected must not include significant changes in operating conditions such as wind speed, turbulence intensity, rotor thrust, or rotational speed. While the original frequency-domain modal analysis algorithms were developed for SISO systems, recent literature also includes extensions to the multivariable case.
One of the first frequency-domain techniques was Peak-Picking or Basic Frequency Domain (BFD), as described in Bendat and Piersol (1980) and Rainieri and Fabbrocino (2014). The user is here required to identify the structural peak in frequency-domain data manually and then fit the damping parameter to achieve the measured intensity and roll-off of this peak. While this algorithm clearly requires significant expert user intervention, it also forms the basis for subsequent algorithms developed in literature, such as the Frequency Domain Decomposition (FDD) technique developed by Brincker et al. (2000b). This method
works on the principle that for a lightly-damped structure in a narrow frequency band of interest around an expected structural mode, the frequency-domain structural response of the system $G_{yy}(j\omega)$ can be approximated as the superposition of a very few modes $k \in \mathrm{Sub}(\omega)$, typically only one or two modes:

$$G_{yy}(j\omega) = \sum_{k \in \mathrm{Sub}(\omega)} \frac{d_k \phi_k \phi_k^T}{j\omega - \lambda_k} + \frac{\bar{d}_k \bar{\phi}_k \bar{\phi}_k^T}{j\omega - \bar{\lambda}_k}, \tag{15}$$

where $d_k$ is a scalar constant; $\phi_k$ and $\lambda_k$ are the mode shape vectors and system eigenvalues respectively. The $\bar{*}$ indicates a
complex conjugate. The FDD method states that the dominant mode shape vector $\phi_i$ in the frequency range of interest can be directly estimated from the SVD of $G_{yy}(j\omega_i)$ at the expected natural frequency $\omega_i$:

$$G_{yy}(j\omega_i) = U_i \Sigma_i U_i^H, \ \hat{\phi}_i = u_{i1}, \tag{16}$$

where $U_i = \begin{bmatrix} u_{i1} & u_{i2} & \cdots & u_{in} \end{bmatrix}$. From the mode shape vector and the FRF, the damping can be estimated. In case two modes are dominant at the frequency $\omega_i$, the FDD algorithm may have an issue in estimating the mode shape of the dominant mode.
However, as for an OWT with a monopile foundation, if the closely-spaced modes (fore-aft and side-side) are orthogonal, then the first two vectors $u_{i1}$ and $u_{i2}$ give an unbiased estimate of both mode shape vectors. Brincker et al. (2000b) demonstrate

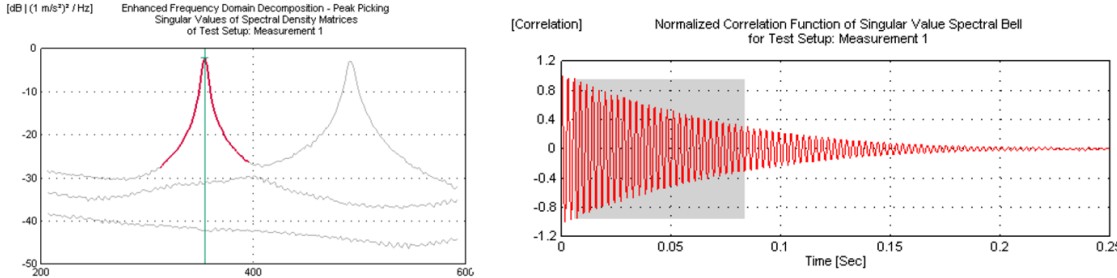

**Figure 6.** Peak-picking and FDD in the frequency domain (left) and the resultant correlation function from the IFT in the time-domain (right) (Jacobsen et al., 2007).

this concept with the modal analysis of a prismatic structure. In general, as described by Rainieri and Fabbrocino (2014), FDD is not able to perform unbiased estimation of closely-spaced modes or harmonics, especially if one of the structural modes is non-dominating in the frequency domain.

Jacobsen et al. (2007) extend this concept in the technique Enhanced Frequency Domain Decomposition (EFDD), which involves an Inverse Fourier Transform (IFT) of the power spectral density function in the narrow frequency band as described above. The damping of the structural mode of interest is then obtained as the logarithmic decrement of the autocorrelation function. This extension is visualised in Fig. 6 and enhances the robustness of the damping estimates obtained using this technique. However, as reported in Rainieri et al. (2010), the damping estimation using EFDD also suffers from reduced 410    accuracy for closely-spaced modes.

While the FDD algorithms described above typically focus on a localised frequency band, the Least Squares Complex Frequency-domain (LSCF) or PolyMAX algorithm developed by Guillaume et al. (1996) fits a dynamic system model to the entire FRF spectrum in a least-squares sense. This algorithm as described by Peeters and Van der Auweraer (2005) works on the assumption that the underlying dynamic system admits a (multivariable) realisation of the form:

$$G(\omega) = B(\omega)A(\omega)^{-1},\tag{17}$$

where the numerator and denominator can be parameterised using the exponential basis functions $\Omega_r(\omega)$ as:

$$B_o(\omega, \beta) = \sum_{r=0}^{n} \Omega_r(\omega)\beta_{or},\tag{18}$$

$$A(\omega, \alpha) = \sum_{r=0}^{n} \Omega_r(\omega)\alpha_r,\tag{19}$$

$$\Omega_r(\omega) = e^{j\omega\Delta tr},\tag{20}$$

where $B_o(\omega)$, $o \in [1,...n_y]$ is each row of the matrix $B$ and $\Delta t$ is the sampling time of the data. Based on the available FRF data, the parameters $(\alpha_r, \beta_{or})$ can be estimated by formulating the estimation error for each output channel at each frequency



as $\varepsilon_o(\omega_k, \beta_o, \alpha)$ in the following manner:

$$\varepsilon_o(\omega_k, \beta_o, \alpha) = w_o(\omega_k)(B_o(\omega_k, \beta_o) - G_y y(\omega_k) A(\omega_k, \alpha)), \tag{21}$$

where $w_o(\omega)$ is a frequency-varying weighting function that can be shaped by the user to force the algorithm to focus on

the frequency range of interest, and ignore the more uncertain parts of the frequency spectrum. Based on the estimation error defined above, the parameters can be estimated by minimising the following suboptimal linear least squares problem:

$$\min_{\alpha_r, \beta_{or}} \sum_{o=1}^{n_y} \sum_{k=1}^{N_f} \mathrm{tr}(\varepsilon_o(\omega_k, \beta_o, \alpha)^H \varepsilon_o(\omega_k, \beta_o, \alpha)), \tag{22}$$

where $N_f$ is the total number of data points available in the FRF $G_{yy}(\omega)$. The parameters $\alpha$ can be obtained by minimising this expression as the nontrivial solution to the following equation:

$$\sum_{o=1}^{n_y} (T_o - S_o^T R_o^{-1} S_o)\alpha = 0, \tag{23}$$

where the terms in the equation are determined as follows:

$$R_o = Re(X_o^H X_o), \ S_o = Re(X_o^H Y_o), \ T_o = Re(Y_o^H Y_o), \tag{24}$$

$$X_o = \begin{bmatrix} w_o(\omega_1) \begin{bmatrix} \Omega_0(\omega_1) & \cdots & \Omega_n(\omega_1) \end{bmatrix} \\ \vdots \\ w_{N_f}(\omega_{N_f}) \begin{bmatrix} \Omega_0(\omega_{N_f}) & \cdots & \Omega_n(\omega_{N_f}) \end{bmatrix} \end{bmatrix}, \tag{25}$$

$$Y_o = \begin{bmatrix} -w_o(\omega_1) \begin{bmatrix} \Omega_0(\omega_1) & \cdots & \Omega_n(\omega_1) \end{bmatrix} \otimes G_{yy}(\omega_1) \\ \vdots \\ -w_{N_f}(\omega_{N_f}) \begin{bmatrix} \Omega_0(\omega_{N_f}) & \cdots & \Omega_n(\omega_{N_f}) \otimes G_{yy}(\omega_{N_f}) \end{bmatrix} \end{bmatrix}. \tag{26}$$

Based on the identified parameters $\alpha$, the modal damping of the system can be determined.

One of the advantages of working in the frequency domain is that the FRF can be smoothed using windowing of the FRF data. It should be noted, however, that the window type should be chosen with caution, as a Hanning window may cause bias in the estimate of the structural damping. Further, with the help of the weighting function in the algorithm, the desired frequency range can be weighted more heavily while higher frequencies corresponding to noise and nonlinearities can be weighted to

a much lighter extent. As a result of these characteristics, it was shown in Peeters and Van der Auweraer (2005) that cleaner identification of modal properties can be achieved by the PolyMAX algorithm as compared to an SSI algorithm without any pre-processing or enhancement techniques. Figure 7 presents a stabilisation diagram showing identified mode against modal order, superimposed on the FRF data used for identification, where it is evident that PolyMAX performs clear and stable identification of the main structural modes of the underlying system for a wide range of chosen model orders.

All modal analysis methods using operational data described above make the assumption that the input excitation is zero mean white noise, but this assumption is no longer valid for operational wind turbine data due to the presence of harmonics, as





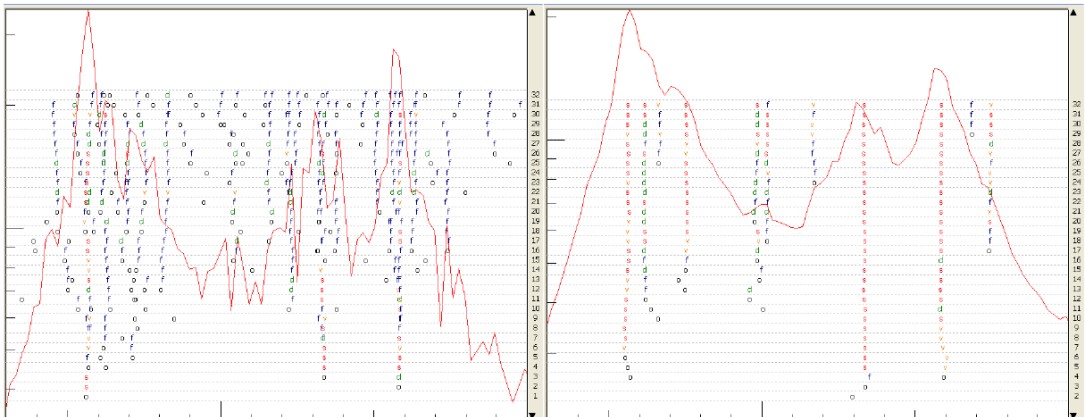

**Figure 7.** Stabilisation diagram using SSI (left) and PolyMAX (right) (Peeters and Van der Auweraer, 2005).

described in Tcherniak et al. (2011). The study by Gasparis (2019) evaluates selected classical OMA algorithms, specifically ERA, SSI, FDD, and LSCF for the estimation of damping from simulated operating wind turbine data. It concludes that although some modes may be identifiable with classical techniques, they are typically not able to distinguish between closely-spaced harmonics and modes. Hence, the subsequent sections will focus on the enhancements made to classical OMA methods to allow high fidelity estimation of OWT structural damping in the presence of harmonics.

## 4 Operational Modal Analysis Algorithms for Data with Known Harmonics

The presence of harmonics in the operational OWT data available can cause an issue for the classical OMA algorithms described in the previous section. For the case where OWT structural modes and harmonics are widely separated, and when the rotor speed is constant over time, harmonics are identified by the OMA algorithms as artificial modes with zero damping. However, if the rotor speed changes over time, it may not be possible to distinguish between structural modes and harmonics in this manner. Further, if the structural modes and harmonics are closely spaced, the harmonic may affect the damping estimation of the structural mode. In recent literature, several authors have attempted to enhance classical OMA algorithms to account for harmonics in measured data. If the harmonics are known, by way of a known and correctly synchronised rotor speed signal, the harmonics can either be removed from the data or directly incorporated into the OMA algorithm.

Removal of known harmonics from OWT data may not be trivial, especially in the case of closely-spaced structural modes and extraneous harmonics. The use of filters may strongly alter the structural damping information contained in measurement data.

To some extent, the drawbacks of harmonic filtering can be alleviated by using Time Synchronous Averaging (TSA) (Peeters et al., 2007), where harmonics can be removed by isolating them from the measurement data. In this method, given the known rotor speed, the response data is resampled into the 'angle' domain using constant angular speed increments. In this domain,



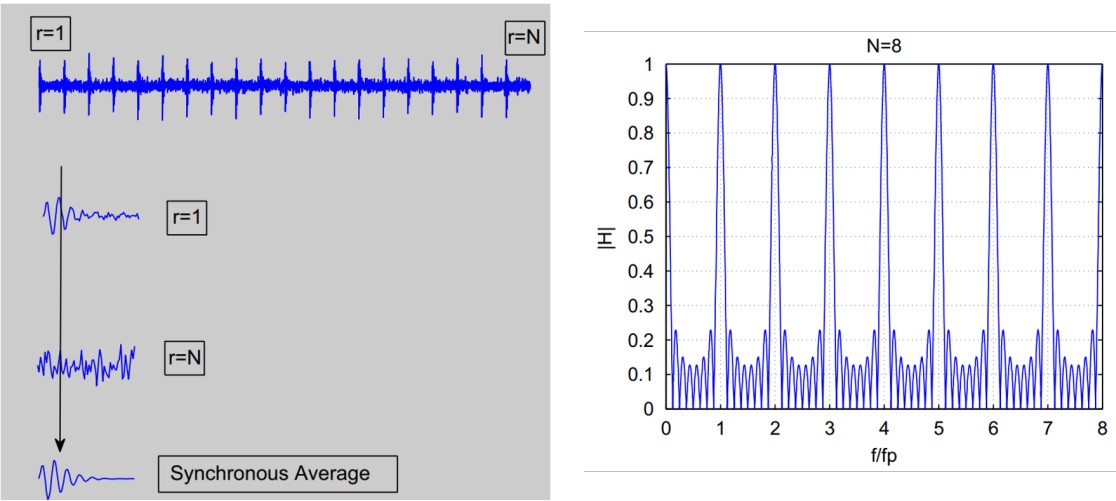

**Figure 8.** Impact of Time Synchronous Averaging (TSA) in the angle domain (left) and the effective frequency response function of TSA (right) (Braun, 2011).

vibration data is sampled over the rotor azimuth instead of time. Synchronous averaging is then performed over multiple angle-domain data series so that harmonic components are clearly separated, and the random components are averaged out, see Fig. 8. Once isolated, the harmonic components can be removed from the raw data in the angle domain, and the signal can be resampled back into the time domain for use with classical OMA algorithms.

TSA requires that the rotor speed is known exactly and stationary, which may not always be the case for OWT measurement data. Another limitation of TSA is that it is applicable only to cases where the spectrum shows sharp harmonic peaks. However, for OWT measurement data, the phenomenon of 'thick-tailed' harmonic peaks described by Tcherniak et al. (2011) limits the applicability of TSA. An application of TSA to damping estimation from operating OWT data has been presented in Manzato et al. (2014), where it is shown that TSA can significantly reduce harmonic peaks in the spectrum of the measurement data, but for the 3P peak and its harmonics, a small peak may remain with significant damping. As an alternative to TSA, OMA algorithms can also be extended to include signal generators, or artificial sinusoidal signals with frequency equal to the known harmonic frequency, directly into the identification procedure.

For instance, Mohanty and Rixen (2004a) extend the LSCE algorithm to include signal generators of the known harmonics present in the measured vibrational data. As motivated above, it is expected that this OMA algorithm will identify the harmonics of known frequency $\omega$ with zero damping as one of the solutions to the linear set of equations. To enforce this behaviour, the authors explicitly include two extra roots that correspond to this harmonic component into Eq. 3 as $x_k = e^{\pm\omega\Delta t}$, where $\Delta t$ is the sampling time of the measured data. This equation is hence modified by the following block of linear equations for each



known harmonic expected to be present in the measurement data:

$$
\quad \begin{bmatrix} 0 & \sin(\omega\Delta t) & \cdots & \sin(\omega(2n-1)\Delta t) \\ 1 & \cos(\omega\Delta t) & \cdots & \cos(\omega(2n-1)\Delta t) \end{bmatrix} \begin{bmatrix} a_0 \\ a_1 \\ \vdots \\ a_{2n-1} \end{bmatrix} = - \begin{bmatrix} \sin(2n\omega\Delta t) \\ \cos(2n\omega\Delta t) \end{bmatrix}. \tag{27}
$$

Herewith, for $m$ known harmonics, the least-squares problem from Eq. 3 becomes:

$$
\begin{bmatrix} R_{2n-1} & \cdots & R_{2n+2m-1} & R_{2n+2m} & \cdots & R_{4n-2} \\ \vdots & \ddots & \vdots & \vdots & \ddots & \vdots \\ R_0 & \cdots & R_{2m-1} & R_{2m} & R_{2n-1} & \\ 0 & \cdots & \sin(\omega_1(2m-1)\Delta t) & \sin(\omega_1 2m\Delta t) & \cdots & \sin(\omega_1(2n-1)\Delta t) \\ 1 & \cdots & \cos(\omega_1(2m-1)\Delta t) & \cos(\omega_1 2m\Delta t) & \cdots & \cos(\omega_1(2n-1)\Delta t) \\ \vdots & \ddots & \vdots & \vdots & \ddots & \vdots \\ 0 & \cdots & \sin(\omega_m(2m-1)\Delta t) & \sin(\omega_m 2m\Delta t) & \cdots & \sin(\omega_m(2n-1)\Delta t) \\ 1 & \cdots & \cos(\omega_m(2m-1)\Delta t) & \cos(\omega_m 2m\Delta t) & \cdots & \cos(\omega_m(2n-1)\Delta t) \end{bmatrix} \begin{bmatrix} a_0 \\ \vdots \\ a_{2m-1} \\ a_{2m} \\ \vdots \\ a_{2n-1} \end{bmatrix} = - \begin{bmatrix} R_{4n-1} \\ \vdots \\ R_{2n} \\ \sin(\omega_1 2n\Delta t) \\ \cos(\omega_1 2n\Delta t) \\ \vdots \\ \sin(\omega_m 2n\Delta t) \\ \cos(\omega_m 2n\Delta t) \end{bmatrix}.
$$

$$\tag{28}$$

It can be seen in the equation above that the impulse response $y_*$ from 3 has been replaced by the cross-correlation $R_*$ following the NExT philosophy. As before, the solution to the equation $\sum_{k=0}^{2n} a_k x^k = 0$ gives the eigenvalues of the solution, including the zero-damping harmonics $x = e^{\pm j\omega_*\Delta t}$ enforced by the additional constraint. Note that, to implement this extension successfully, the harmonics should be stationary and known a priori. Mohanty and Rixen (2004a) show that a minor deviation from the exact value of the harmonic can already produce incorrect identification results. A similar extension of the ERA algorithm has been presented in Mohanty and Rixen (2006), where the harmonic solution $x_k = e^{\pm j\omega_*\Delta t}$ is included in the block Hankel matrix. The same authors have also extended the ITD and the SSTD algorithms to include known harmonics in a similar manner in Mohanty and Rixen (2004b) and Mohanty and Rixen (2004c), respectively. As in the modified LSCE algorithm, these modified algorithms are able to identify structural modes more accurately, as long as the harmonics included in the algorithm are stationary and accurately known a priori.

Similarly, for harmonics in the input spectrum, the SSI approach was modified to the Harmonic Modification-SSI (HM-SSI) approach by Dong et al. (2014). This algorithm incorporates harmonics in the procedure in a manner similar to the modified LSCE, namely, by extending the Hankel data matrix to include the known harmonic signals. HM-SSI can successfully distinguish between structural modes and harmonics, where classical SSI fails, as shown in Fig. 9.

For all algorithms described above, it is necessary to extend the Hankel data matrix to include harmonics. Accurate a priori knowledge of harmonics is required, and the algorithms degenerate to the classical algorithms if such information is not available. When harmonics are not known a priori, statistical techniques may be needed and are discussed in the next section.



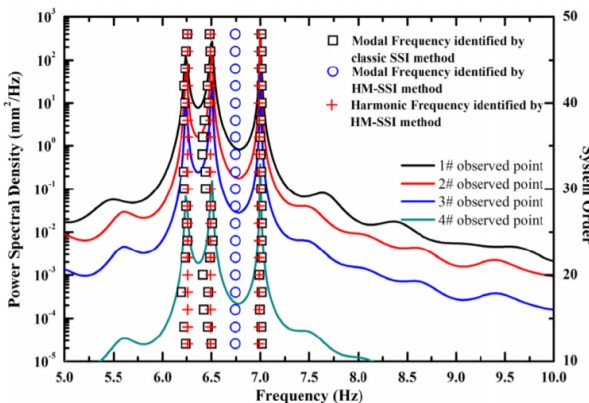

**Figure 9.** Stability diagram of system excited by white noise and three harmonic components. HM-SSI identified the harmonics as false modes, while SSI identifies them as structural modes. The true modal frequency is found by HM-SSI (Dong et al., 2014).

## 5   Operational Modal Analysis Algorithms for Data with Unknown Harmonics

If harmonics of unknown frequency are present in the data, classical OMA algorithms identify both structural modes and harmonics as system modes. Harmonics can typically be identified as the estimated modes with zero damping. However, due to variations in OWT rotor speed, turbine harmonics might be more difficult to identify as such in the measurement spectrum, especially in below-rated operation. Statistical indicators may be used for differentiating harmonics from structural modes. This section summarises three such indicators: the signal cepstrum, the Probability Density Function (PDF), and kurtosis.

### 5.1   Cepstrum

The concept of the cepstrum has been described by Randall and Hee (1982) and applied for OMA by Randall et al. (2012). The original definition of the cepstrum was as a 'spectrum of a spectrum'. Harmonics appear as periodic peaks in the typical spectrum, while the structural peaks are broadly and randomly distributed. If the cepstrum of the signal is derived, the harmonics are converted into clearly identifiable cepstral peaks, the so-called 'rahmonics'. The structural modes are no longer visible in the cepstrum and hence isolation of harmonics is achieved.

To ensure that the cepstral transformation is fully reversible and the original spectrum and time series can be retrieved from the cepstrum, the cepstrum has been redefined by Randall et al. (2012) as the inverse Fourier transform of the *logarithm* of the power spectrum of the signal. The cepstrum $C_c(\tau)$ of a signal $x(t)$ with spectrum $X(\omega)$ is hence defined as:

$$X(\omega) = \mathcal{F}(x(t)), \tag{29}$$

$$C_c(\tau) = \mathcal{F}^{-1}(\log(X(\omega))), \tag{30}$$

where $\mathcal{F}$ is the Fourier operator.





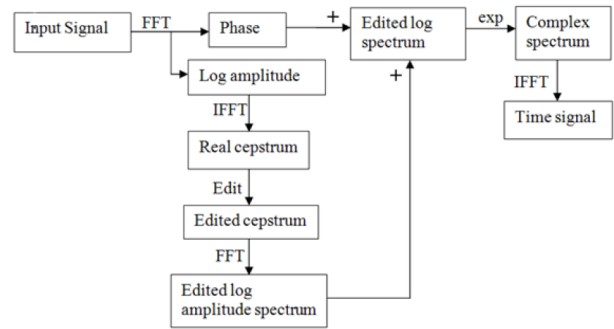

**Figure 10.** Schematic diagram of the cepstral method for removing harmonics from time-domain signals (Randall et al., 2012).

.

The concept of cepstral rahmonics corresponding to data harmonics remains valid in this new definition. Once identified, the rahmonics can be edited out. The original data harmonics are then eliminated if a time-domain signal is reconstructed from this edited cepstrum. This process is depicted schematically in Fig. 10, and a practical example is shown in Fig. 11.

Apart from selectively editing rahmonics in the signal cepstrum, Randall et al. (2012) also describe the use of a lowpass 'lifter' (filter in the cepstral domain). This implies the removal of the higher end of the signal cepstrum using a rectangular or exponential window, as this part of the signal is expected to correspond to noise. One advantage of this approach is the ease of automation since specific rahmonics do not need to be identified and edited. It should be noted that exponential liftering alters the damping of the modes in the signal, and a post-processing correction is required to the damping estimates. A combination of liftering and rahmonic editing can increase the quality of damping estimation.

Automated low-pass liftering of OWT measurement data has been described in Manzato et al. (2014). It is shown in this case that if sufficient spacing is not present in the original data between the structural mode and the harmonic, cepstral liftering can strongly alter the structural information content of the signal and lead to poor damping estimates. However, the combination of notch liftering of rahmonics and subsequent low-pass liftering of OWT measurement data in Manzato et al. (2013) combined with a PolyMAX modal identification step shows significant improvement in identification fidelity. Specifically, the cepstral preprocessing step increases the number of structural frequencies identified by the PolyMAX algorithm, as described by Peeters et al. (2004). Furthermore, harmonics are no longer identified as structural modes by this algorithm.

In the references above, the cepstrum makes use of the periodicity of unknown harmonics for their removal. Alternative approaches for identifying unknown harmonics exploit the difference in the statistical distribution of the structural response and harmonic response to perform the pre-processing harmonic isolation, as described in the next subsections.

## 5.2 Probability Density Function (PDF)

Once the system poles have been estimated using a classical OMA algorithm, the structural modes can be distinguished from harmonics based on the PDF. The differentiation is based on the difference between the shape of the PDF of a stochastic

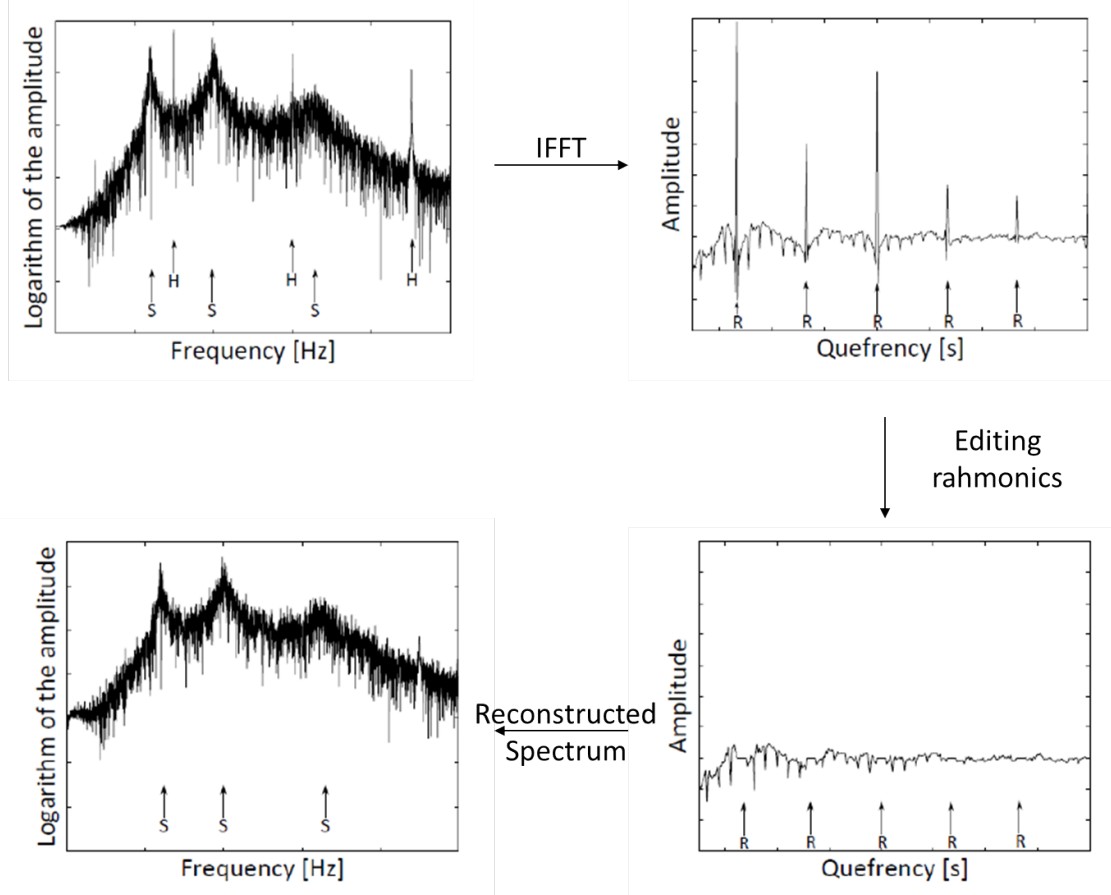

**Figure 11.** Cepstral method for removal of harmonics. The harmonics are indicated by H, the structural modes by S, and the rahmonics in the cepstral domain by R (Motte et al., 2015).

structural response and that of a harmonic excitation, as depicted in Fig. 12. As described in Brincker et al. (2000a), the structural response PDF is approximately Gaussian, while the ideal harmonic PDF $g(y)$ is:

$$g(y) = (\pi \cos(\sin^{-1} \frac{y}{a}))^{-1}, \tag{31}$$

where $a$ is the amplitude of the harmonic signal. This insight can be directly applied in conjunction with classical frequency domain identification techniques such as FDD to separate structural modes from harmonics. Determining the PDF of the signal treated with a narrow band-pass filter around the spectrum peak of interest directly gives information on whether the peak corresponds to a structural mode or a harmonic.

It has been shown in Motte et al. (2015), that the harmonics from operational turbine data may not fully conform to the shape shown in Fig. 12. Further, because the evaluation of the PDF is to some extent qualitative, automation of this procedure is considered to be difficult.





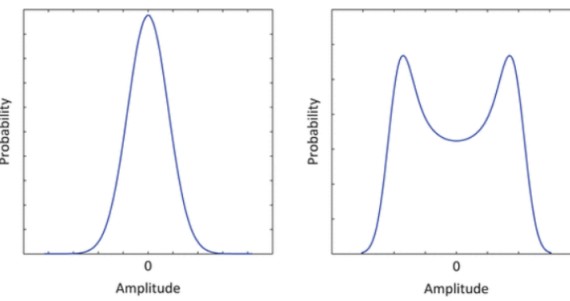

**Figure 12.** Comparison between the different shapes of a normalised PDF of a structural mode (left) and a harmonic (right) (Motte et al., 2015).

## 5.3 Kurtosis

Kurtosis has been proposed as a numerical indicator for distinguishing between the PDF of a structural mode and a harmonic. Kurtosis $\gamma$ is defined as the fourth central moment of stochastic variable $x$:

$$\gamma(x|\mu,\sigma) = \frac{E[(x-\mu)^4]}{\sigma^4}, \tag{32}$$

where $\mu$ and $\sigma$ are the mean and standard deviation of $x$, respectively. The kurtosis of the signal can be calculated, after band-pass filtering the signal around the spectrum peak of interest. If the band-pass filtered signal corresponds to a structural mode, it shows a normal PDF as in the above section, and the kurtosis value will be close to 3. On the other hand, for pure harmonic signals of zero mean and unit variance, the kurtosis is close to 1.5. Based on this difference in kurtosis values, identified modes can be classified as structural modes and harmonics.

This approach has been used to identify and edit harmonics from the frequency spectrum before applying the frequency-domain EFDD algorithm by Jacobsen et al. (2007) for the identification of structural modes. While this approach shows better results compared to classical EFDD, the performance deteriorates as the harmonics and structural modes get closer. Editing harmonics is also a nontrivial exercise - while the authors use linear interpolation to replace the edited harmonics, polynomial interpolation has also been suggested for better spectral quality of the input data to the EFDD algorithm.

Statistical indicators like kurtosis can also be used to isolate harmonics in SSI algorithms. Once the harmonics are identified, the damping estimate can be enhanced using an algorithm like the Kalman Filter-based SSI (KF-SSI) technique developed by Greś et al. (2021). Here, a Kalman filter is used to obtain an estimate of the harmonic component of the system state. Subsequently, as in several classical subspace identification algorithms like MOESP, the row space of the raw output data is projected onto the complement of the row space of the harmonic subsignal estimate, to remove the influence of harmonic perturbation. A refined damping estimate can now be obtained by applying classical SSI techniques to the projected data.

Greś et al. (2021) demonstrate the advantage of KF-SSI over classical SSI techniques experimentally with measurements from a vibrating plate, excited with a harmonic signal with its frequency close to the first structural frequency of the plate. As



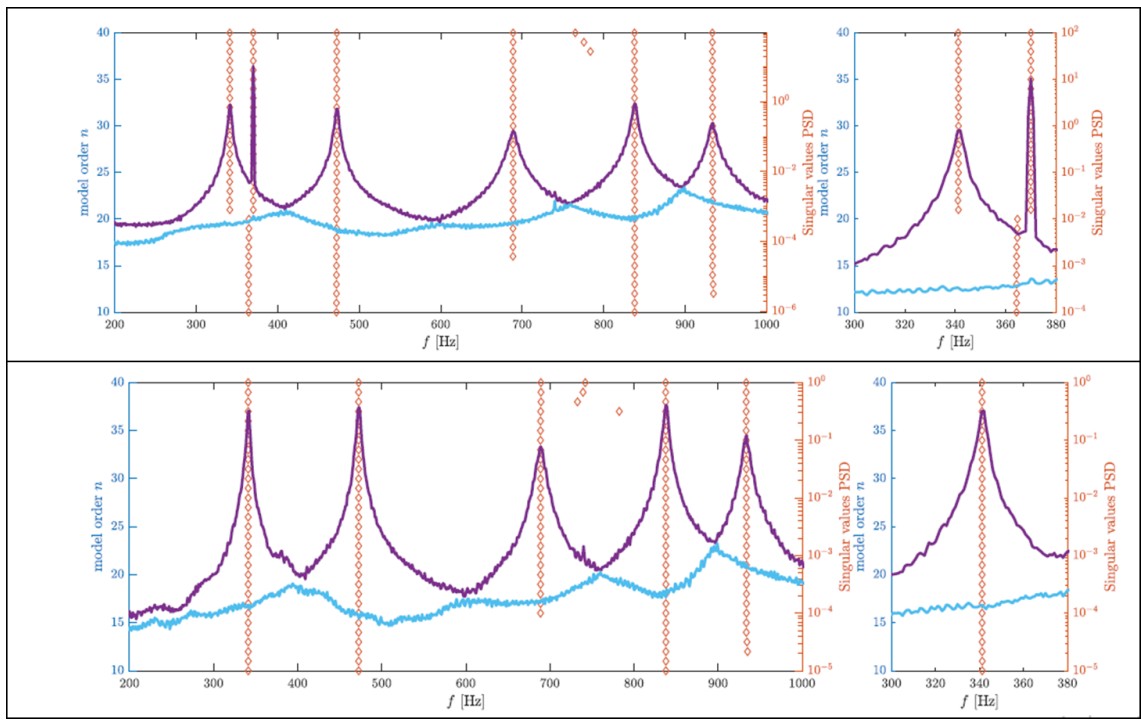

**Figure 13.** Stabilisation diagram of the signal after applying classical SSI (top) and KF-SSI (bottom). For orders lower than 20, SSI fails to distinguish between the structural mode and harmonic and presents a merged version, while KF-SSI exclusively identifies the structural mode (Greś et al., 2021).

seen in Fig. 13, classical SSI is unable to distinguish between the structural frequency and the harmonic for model order below 20, while KF-SSI is able to identify the structural frequency also at low model orders.

While the extensions above reduce the sensitivity of classical OMA algorithms to the presence of harmonics, there also exist
algorithms based on transmissibility that are by construction insensitive to rotor harmonics. These will be reviewed in the next section.

## 6   Transmissibility-based Algorithms

Transmissibility-based OMA algorithms identify the structural parameters of the OWT in a manner completely independent of the load spectrum. This recently-developed family of algorithms can be exemplified by the Transmissibility-based Opera-
tional Modal Analysis (TOMA) algorithm introduced by Devriendt and Guillaume (2007). These algorithms are based on the transmissibility function $T_{ij}$, defined as the ratio between two structural responses $X_i(\omega)$ at location $i$ and $X_j(\omega)$ at location $j$, under loading condition $k$:

$$T_{ij}^k = \frac{X_i(\omega)}{X_j(\omega)} = \frac{H_{ik}(\omega)}{H_{jk}(\omega)}. \tag{33}$$




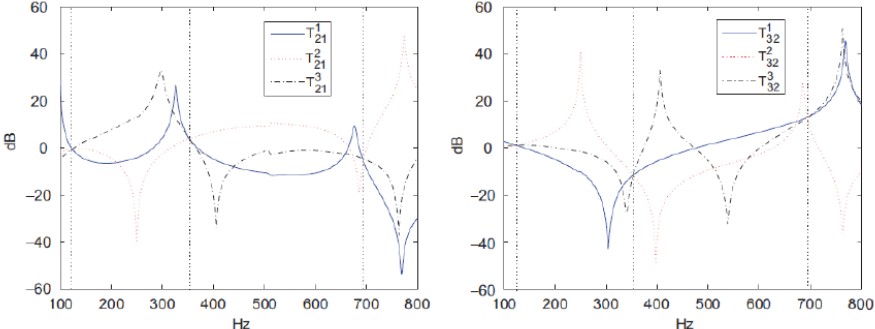

**Figure 14.** Transmissibility functions for three different loading conditions $T^k$, for outputs 1 and 2 (left) and outputs 2 and 3 (right). The systems poles lie at the intersection of the functions indicated by vertical dashed lines (Devriendt and Guillaume, 2007).

Here, $H_{*k}(\omega)$ is the (unknown) transfer function from the unmeasured force inputs to the responses measured at $*$. Transmissibility functions do not require a strictly white noise input force spectrum (Devriendt and Guillaume, 2008), and are hence ideal for OWT damping estimation. The points of intersection of multiple transmissibility functions correspond to the poles of the underlying dynamic system, see Fig. 14. Hence, the poles of the system are the same as those of $\Delta^{-1}T_{ij}^{k\ell}(s)$ defined as:

$$\Delta^{-1}T_{ij}^{k\ell}(s) = \frac{1}{T_{ij}^k - T_{ij}^\ell}. \tag{34}$$

Since the frequency response of this system can be constructed from measurement data, a classical frequency-domain OMA method can subsequently be used for damping estimation using the transmissibility concept. The concept of transmissibility has been extended to multivariable systems by Maia et al. (2001), and TOMA for multivariable systems has been presented by Devriendt et al. (2010). As long as persistency of excitation is ensured, damping estimation can be performed independent of the presence of harmonics in the measurement data. However, to ensure a well-conditioned intersection of transmissibility functions, the loading conditions used for generating each function should be sufficiently different, as described by Weijtjens et al. (2014a). The authors present a polyreference version of TOMA (pTOMA), which uses a polynomial parametric model to perform modal identification based on the frequency response function of $\Delta^{-1}T_{ij}^{k\ell}(s)$ for a limited number of loading conditions. Theoretically, the authors prove that damping estimation can be performed based on the data from only two different loading conditions, but identification fidelity improves when data from multiple different loading conditions is available.

Since multiple datasets for different loading conditions may not be trivial to obtain for OWT data, Weijtjens et al. (2014b) introduce Time-Varying TOMA analysis (TV-TOMA), which uses time-varying transmissibility functions. Here, the assumption of stationarity of data is relaxed, and the loading is permitted to be varying continuously. It is then possible to use different time sample periods from the same dataset to determine system damping using transmissibility functions, as shown in Fig. 15. This significantly reduces the overall amount of measurement data required for transmissibility-based damping estimation.

An extension of frequency-domain identification using the Power Spectral Density Transmissibility (PSDT) concept was developed by Yan and Ren (2011) to relax the TOMA requirement of multiple loading conditions. The PSDT is defined in




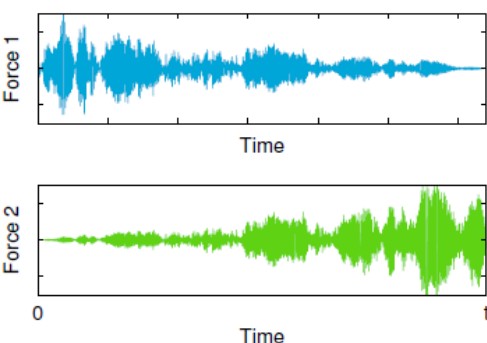
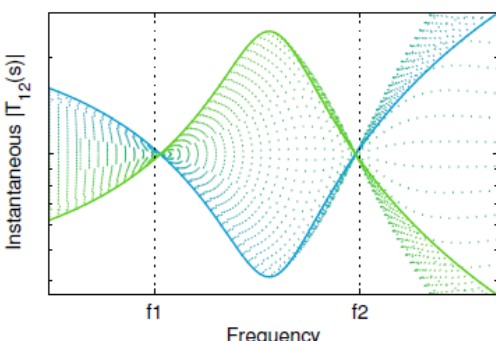

**Figure 15.** Instantaneous transmissibility functions $T_{12}$ (right) shown for the case where the input location of the load varies over time from position 1 to 2 (left). Instantaneous values of $T_{12}$ show the same property as time-invariant transmissibilities in that they converge at the system poles (Weijtjens et al., 2014b).

a manner similar to the definition of the transmissibility (Eq. 33), using the cross Power Spectral Density (PSD) $S_{x_z x_u}(j\omega)$ between the measured output $x_u$ and the reference output $x_z$ which admits the following modal description:

$$
\begin{aligned}
S_{x_z x_u}(j\omega) = \varphi_{z1}^* &\left[ \sum_{r=1}^{N}\sum_{s=1}^{N} H_{1f_r}(j\omega)^* S_{f_r f_s}(j\omega) H_{1f_s}(j\omega)\varphi_{u1} + \cdots + \sum_{r=1}^{N}\sum_{s=1}^{N} H_{2f_r}(j\omega)^* S_{f_r f_s}(j\omega) H_{n_m f_s}(j\omega)\varphi_{un_m} \right] + \cdots \\
+ \varphi_{z2}^* &\left[ \sum_{r=1}^{N}\sum_{s=1}^{N} H_{2f_r}(j\omega)^* S_{f_r f_s}(j\omega) H_{1f_s}(j\omega)\varphi_{u1} + \cdots + \sum_{r=1}^{N}\sum_{s=1}^{N} H_{2f_r}(j\omega)^* S_{f_r f_s}(j\omega) H_{n_m f_s}(j\omega)\varphi_{un_m} \right] + \cdots \\
+ \varphi_{zn_m}^* &\left[ \sum_{r=1}^{N}\sum_{s=1}^{N} H_{n_m f_r}(j\omega)^* S_{f_r f_s}(j\omega) H_{1f_s}(j\omega)\varphi_{u1} + \cdots + \sum_{r=1}^{N}\sum_{s=1}^{N} H_{n_m f_r}(j\omega)^* S_{f_r f_s}(j\omega) H_{n_m f_s}(j\omega)\varphi_{un_m} \right],
\end{aligned}
$$

where $\varphi_{*\ell}$ are the mode shape vectors of mode $\ell$, and $H_{\ell f_r}$ is the $\ell^{\text{th}}$ component of the transfer function between the unknown input force $f_r$ and the output. The number of modes is $n_m$, while the number of input forces is $N$. The PSDT $T_{x_u x_k}^{x_z}(j\omega)$ is defined in terms of two cross-PSD's $S_{x_z x_u}(j\omega)$ and $S_{x_z x_k}(j\omega)$ for the reference output $x_z$ as:

$$
T_{x_u x_k}^{x_z} = S_{x_z x_u}(j\omega)^T (S_{x_z x_k}(j\omega)^T)^\dagger, \tag{35}
$$

where † represents the Moore-Penrose pseudo-inverse for a rectangular matrix. It can be observed that the PSDT of the system can be constructed purely based on measurement data.

Like the transmissibility, the PSDT is also independent of the input loading spectrum and the reference outputs at the system poles. The system poles are now located at the intersection of PSDT's with different reference outputs. As such, this approach does not require different loading conditions for the estimation of the structural parameters, as long as distinct reference outputs are available, unlike the TOMA algorithm described above.

PSDT was improved to an SVD-based algorithm (PSDTM-SVD) by Araújo and Laier (2014), to enhance its numerical robustness. These PSDT algorithms primarily focus on the identification of structural frequencies. To determine the structural



damping of the system, it is required to perform model estimation of $(T^{x_z}_{x_u x_{k1}} - T^{x_z}_{x_u x_{k2}})^{-1}$ using a classical (frequency-domain) algorithm like LSCF, as incorporated into the Enhanced PSDT (EPSDT) algorithm by Yan and Ren (2015). The algorithm has been extended to multivariable system identification by Araújo and Laier (2015).

Like all transmissibility-based algorithms, the PSDT approaches are insensitive to data contamination caused by the presence of harmonics in the OWT measurement data. All OMA approaches described in the foregoing sections are now compared based on their suitability for OWT damping estimation in the next section.

## 7 Algorithm suitability for OWT identification

For the express purpose of OWT damping estimation based on measurement data, all algorithms described in the above sections can be evaluated in terms of their suitability. It is expected that the most suitable algorithm shall satisfy the following suitability criteria to the largest extent possible:

1. The algorithm shall be able to perform identification based on output-only measurement data.

2. The algorithm shall be able to estimate OWT damping accurately using finite durations of measurement data.

3. The algorithm shall be able to distinguish between closely-spaced structural modes, especially if they are orthogonal.

4. The computational complexity of the algorithm shall be as minimal as possible.

5. The algorithm shall be able to handle harmonics present in operational OWT measurement data.

6. The algorithm shall be able to identify structural modes accurately even when they are located close to harmonics in the spectrum of the measurement data.

7. The algorithm shall be able to handle the non-stationarity of harmonics, which predominantly characterises the harmonics present in operational OWT data in ordinary turbulent wind climates.

8. The algorithm shall be able to handle high harmonic energy content since harmonics in OWT data spectra are often of the same magnitude as structural modes.

9. The algorithm shall preferably not require a priori knowledge of harmonics, since accurate and perfectly synchronised measurements of OWT rotor speed may not be available.

The first four criteria are desirable for any robust OMA algorithm. From the fifth criterion onwards, attention is directed primarily to the harmonics that typically contaminate OWT measurement data. The algorithms described in the foregoing sections may or may not meet all nine of the suitability criteria here defined. However, depending on the exact nature of instrumentation of the turbine and availability of data in different operating conditions, some of the criteria above may be relaxed in regards to the suitability of the OMA algorithms for OWT damping estimation.




Table 2 evaluates each of the algorithms described in this paper according to the suitability criteria defined above. The ✓ and ✗ indicate whether or not the criterion is satisfied, while the ~ indicates that a firm conclusion cannot be drawn based on the current literature review. As described in the foregoing sections, the classical OMA algorithms are typically not able to handle harmonic contamination of data, although the harmonic is easier for an analyst to exclude in the case of frequency

domain techniques like FDD/EFDD. When the harmonic or OWT rotation speed is known accurately, the modified family of algorithms or HM-SSI can be used by including the harmonics directly into data Hankel matrices.

For the case where harmonics are unknown, statistical measures can be used to isolate harmonic data, where kurtosis and cepstral techniques are easier to automate and hence preferred over the PDF approach. KF-SSI can make use of any of these techniques to orthogonally remove the harmonic contamination. It is hence able to satisfy nearly all suitability criteria, apart from

665 the ability to deal with non-stationary harmonics. The presence of harmonics theoretically does not affect the transmissibility-based algorithms, however, literature results have shown insufficient identification accuracy in the case of high harmonic energy in the measurement data.

In conclusion, based on the above table, a selection of different algorithms may be adopted by the user as per the availability and characteristics of the (operational) measurement data, for the damping estimation of the OWT structure.

## 8   Discussion and practical implementation issues

For the OMA algorithms defined above, one of the post-processing approaches common to all is the 'stabilisation diagram', which plots all 'stable' modes identified (Reynders et al., 2012). This approach often uses heuristics to classify the stability of poles, and the final damping estimates are strongly influenced by this classification. A rigorous mathematical interpretation of such a diagram is missing, as is the proof that such a diagram will converge asymptotically to the true OWT damping.

Classical time-domain OMA methods such as ERA-NExT can be used to perform OWT damping estimation when the turbine is idling. However, the level of excitation is low, and the total OWT idling lifetime spent is a small fraction of its total lifetime. As such, it is more desirable to consider damping estimation from operational measurement data.

Operational measurement data is typically contaminated by strong harmonics that may be spaced close to the structural modes and may hence impede identification using classical time-domain techniques. Classical frequency-domain techniques,

both local (e.g. EFDD) and global (e.g. LSCF) generally fare better, in that the harmonic is easier to pick out and separate from the structural modes. Although global frequency-domain techniques are easier to automate than local techniques, both suffer from the drawback that manual intervention may be needed to eliminate harmonic peaks, and such spectral editing can strongly alter the information content of nearby structural modes.

For the case where the harmonic content is known, it can be accounted for by including this information in the methods like

HM-SSI. The drawback here is that unless the harmonics are known with high precision, these methods degenerate to their classical counterparts. For OWT data, with typically non-stationary harmonics, these approaches may not deliver good results unless the wind/wave climate is exceptionally stable.



**Table 2.** Algorithm suitability for damping estimation from (operational) OWT data

| Algorithm | Category | Reference | Suitability Criterion | | | | | | | | |
|---|---|---|---|---|---|---|---|---|---|---|---|
| | | | 1 | 2 | 3 | 4 | 5 | 6 | 7 | 8 | 9 |
| ITD | Classical (Time Domain) | Ibrahim (1973) | ✗ | - | - | - | - | - | - | - | - |
| (p)LSCE | Classical (Time Domain) | Brown et al. (1979) | ✗ | - | - | - | - | - | - | - | - |
| SSTD | Classical (Time Domain) | Zaghlool (1980) | ✗ | - | - | - | - | - | - | - | - |
| ERA | Classical (Time Domain) | Juang and Pappa (1985) | ✗ | - | - | - | - | - | - | - | - |
| SSI | Classical (Time Domain) | Van Overschee and De Moor (1991) | ✓ | ✓ | ✓† | ~ | ✗ | ✗ | ✗ | - | - |
| ITD-NExT | Classical (Time Domain) | James et al. (1995) | ✓ | ✓ | ✗* | ~ | ✗ | ✗ | ✗ | - | - |
| (p)LSCE-NExT | Classical (Time Domain) | James et al. (1995) | ✓ | ✓ | ? | ~ | ✗ | ✗ | ✗ | - | - |
| ERA-NExT | Classical (Time Domain) | James et al. (1995) | ✓ | ✓ | ✗‡ | ~ | ✗ | ✗ | ✗ | - | - |
| FDD | Classical (Freq. Domain) | Brincker et al. (2000b) | ✓ | ✗ | ✗† | ✓ | ✓ | ✗ | ✓ | ✗ | ✗ |
| EFDD | Classical (Freq. Domain) | Jacobsen et al. (2007) | ✓ | ✓ | ~§ | ✓ | ✓ | ✗ | ✓ | ✗ | ✗ |
| LSCF | Classical (Freq. Domain) | Guillaume et al. (1996) | ✓ | ✓ | ✗† | ✓ | ✗ | ✗ | ✗ | - | - |
| PolyMAX | Classical (Freq. Domain) | Peeters and Van der Auweraer (2005) | ✓ | ✓ | ✓† | ✓ | ✗ | ✗ | ✗ | - | - |
| Modified ITD | Known Harmonics | Mohanty and Rixen (2004b) | ✓ | ✓ | ✗* | ~ | ✓ | ✓ | ✗ | ✗ | ✗ |
| Modified (p)LSCE | Known Harmonics | Mohanty and Rixen (2004a) | ✓ | ✓ | ? | ~ | ✓ | ✓ | ✗ | ? | ✗ |
| Modified SSTD | Known Harmonics | Mohanty and Rixen (2004c) | ✓ | ✓ | ✗* | ~ | ✓ | ✓ | ✗ | ? | ✗ |
| TSA | Known Harmonics | Peeters et al. (2007) | ✓ | - | - | ~ | ✓ | ✓ | ✗ | ? | ✗ |
| HM-SSI | Known Harmonics | Dong et al. (2014) | ✓ | ✓ | ✓† | ~ | ✓ | ✓ | ✗ | ✓ | ✗ |
| PDF | Unknown Harmonics | Brincker et al. (2000a) | ✓ | ✗ | ✗† | ✓ | ✓ | ✗ | ✓ | ✗ | ✓ |
| Kurtosis | Unknown Harmonics | Jacobsen et al. (2007) | ✓ | ✓ | ~§ | ✓ | ✓ | ✗ | ✓ | ✗ | ✓ |
| Cepstral Editing | Unknown Harmonics | Randall et al. (2012) | ✓ | - | - | ✓ | ✓ | ✗ | ✓ | ✓ | ✓ |
| KF-SSI | Unknown Harmonics | Greś et al. (2021) | ✓ | ✓ | ✓† | ~ | ✓ | ✓ | ✗ | ✓ | ✓ |
| TOMA | Transmissibility-Based | Devriendt and Guillaume (2007) | ✓ | ✓ | ~¶ | ✓ | ✓ | ✓ | ✗ | ? | ✓ |
| pTOMA | Transmissibility-Based | Weijtjens et al. (2014a) | ✓ | ✓ | ~¶ | ✓ | ✓ | ✓ | ✓ | ? | ✓ |
| TV-TOMA | Transmissibility-Based | Weijtjens et al. (2014b) | ✓ | ✓ | ~¶ | ✓ | ✓ | ✓ | ✓ | ? | ✓ |
| PSDT | Transmissibility-Based | Yan and Ren (2011) | ✓ | ✗ | ~¶ | ✓ | ✓ | ✓ | ✓ | ? | ✓ |
| EPSDT | Transmissibility-Based | Yan and Ren (2015) | ✓ | ✓ | ~¶ | ✓ | ✓ | ✓ | ✓ | ? | ✓ |
| Multivariable EPSDT | Transmissibility-Based | Araújo and Laier (2015) | ✓ | ✓ | ~¶ | ✓ | ✓ | ✓ | ✓ | ? | ✓ |

*Algorithm evaluated by Malekjafarian et al. (2010).

§Algorithm evaluated by Rainieri et al. (2010), and † by Rainieri and Fabbrocino (2014).

‡Algorithm evaluated by Bajric et al. (2015).

¶Algorithm evaluated by Araújo et al. (2018).

Statistical alternatives exist to estimate unknown harmonic content, for instance, based on the PDF or kurtosis of the (filtered) data. The non-stationarity of OWT harmonics may still render these steps non-trivial and may warrant significant manual inter-

 

vention for successful harmonic isolation. Potential future work lies in the direction of combining uncertain and non-stationary rotor speed measurements with statistical techniques for robust and automated elimination of harmonic contamination of OWT data using OMA methods such as KF-SSI.

The transmissibility-based algorithms were developed specifically for their insensitivity to harmonics. However, unless the loading conditions or measurement channels are sufficiently separated, they may suffer from a lack of well-conditioned transmissibility measures, leading to increased uncertainty in the final damping estimates as compared to SSI (Yan et al., 2019).

It should be noted that apart from the classical SSI techniques, very few of the above-mentioned techniques have a formal proof of convergence to the true underlying structural parameters. However, it has been shown with several studies that, given sufficient data, a good estimate of the OWT structural damping may be obtained under certain limiting conditions of stationarity and sufficient persistency of excitation. Future work would need to focus on the relaxation of the condition of stationarity, which severely limits the amount of data usable for identification.

## 9 Conclusions

Significant attention has been devoted in recent literature to the development of OMA methods for the estimation of OWT structural damping, a property that has a significant influence on turbine loads but remains difficult to quantify using first-principles approaches. To validate these damping values, one has to resort to the identification of OWT damping based on measurement data, typically obtained from motion or loads sensors located on the turbine structure. Due to the lack or inadequacy of input data, it is also requires to focus on output-only OMA techniques for damping estimation from OWT measurement data.

Depending on the exact OWT configuration and measurement data available, a choice can be made out of several different OMA techniques, based on the novel suitability criteria table developed in this paper. For increased identification fidelity, a multidisciplinary design of experiment is essential in order to define the optimal choice of a minimum number of sensors and a minimum number of datasets for determining the solution of the damping estimation problem with minimum uncertainty. For a modern OWT, a consistent benchmark comparison of all OMA techniques for damping estimation would also prove to be an invaluable starting point for OWT structural engineers so that these high-quality damping estimates could subsequently be used for the structural design of the increasingly cost-competitive new generation of offshore wind turbines.

*Author contributions.* This document forms part of the work for the thesis of the first author. All other authors have provided practical and theoretical guidance for the development of the material for this paper.

*Competing interests.* The authors confirm that no competing interests are present.



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
