# Peer review of "Damping Identification of Offshore Wind Turbines using Operational Modal Analysis: A Review"

_Wind Energy Science, 2021_

## Author Comment (AC2)

| | |
|---|---|
| Date | October 27, 2021 |
| Our reference | n/a |
| Your reference | n/a |
| Contact person | S.T. Navalkar |
| Telephone/fax | +31 (0)15 27 86707 / n/a |
| E-mail | Sachin.Navalkar@siemensgamesa.com |
| Subject | Response to referees |

**Delft University of Technology**

Delft Center for Systems and Control

Address
Mekelweg 2 (3ME building)
2628 CD Delft
The Netherlands

www.dcsc.tudelft.nl

Anonymous Referee #1, Anonymous Referee #2
*Referees, Wind Energy Science*

Dear Referees,

First of all, the authors would like to thank the referees for their positive and constructive feedback. We believe that the comments have helped us to improve the quality of the paper. In our attempt to account for the comments, we have thoroughly revised different aspects of the paper. The objective of this document is to respond to the points raised by the referees and to provide a detailed overview of the changes made to the paper. The document consists of two sections where we will respond to the review report provided by each of the referees.

Yours sincerely,

Sachin Navalkar
Mees van Vondelen
Alexandros Iliopoulos
Daan van der Hoek
Jan-Willem van Wingerden

Enclosure(s): Response to comments of Anonymous Referee #1
Response to comments of Anonymous Referee #2

**Response to comments of Anonymous Referee #1**

- As a negative point, it can be said that the fact that the work is focused in offshore wind turbines it is not always clear, which originates from the structural similarity to onshore wind turbines. The text would benefit from a small mention to the onshore counterparts, though this is not mandatory.
  A clarifying mention was made to the onshore counterparts in the introduction.

**Response to comments of Anonymous Referee #2**

- The reviewer thinks that the effort the authors spent in their introduction to justify the need for more accurate damping estimation would better be directed to the damping estimation during the real lifetime of the structure. Continuous estimation of damping along with other structural properties would enable continuous updating of the lifetime predictions. If the initial conservative damping assumptions were replaced continuously by more realistic damping estimates longer lifetimes associated with economic benefits can be expected. However, it must be kept in mind that the accuracy of lifetime predictions depends on the length of prediction times and does not only depend on the estimated structural properties but, for example, also on the implemented inspection philosophy.
  The authors agree that the main benefit from operational damping estimation can be gained with improving estimations for lifetime predictions rather than optimizing structural design during the design phase. The authors also believe that the successful estimation of structural damping in operational projects will help in a better understanding of the phenomenon, in turn leading to more accurate damping assumptions in the design phase. It is expected that such an increase in the accuracy of damping models will be accompanied by a reduction in the level of conservatism currently demanded in wind turbine design. A modification to this motivation was made throughout the entire paper.

- Is the suitability criterion fulfilment in table 2 reported from literature or is it derived from the authors' own judgement?
  Although the authors evaluated several of the considered algorithms in a practical case, the conclusions in table 2 are drawn from current literature only. A clarifying sentence has been added in the introduction to emphasize this.

- Did the present authors evaluate one or more algorithms by their own software implementations? The authors evaluated the SSI, KF-SSI, PolyMAX, Enhanced PSDT, LSCE, Cepstrum editing and Modified LSCE algorithms on experimental and simulation data from an operational offshore wind turbine. The results from this study will be presented in a future publication.

- Looking on the notation used for the equations the authors should improve the definition for the indices. For example, in Eq.(1) the sample point k and indices t1 are not explained. Index t1 is not unique on the left hand and the right hand side of the equation. Other equations should be reviewed accordingly.
The missing definitions of indices have been added.

---

## Author Response (AR2)

| | |
|---|---|
| Date | December 3, 2021 |
| Our reference | n/a |
| Your reference | n/a |
| Contact person | S.T. Navalkar |
| Telephone/fax | +31 (0)15 27 86707 / n/a |
| E-mail | Sachin.Navalkar@siemensgamesa.com |
| Subject | Response to referee |

**Delft University of Technology**

Delft Center for Systems and Control

Address
Mekelweg 2 (3ME building)
2628 CD Delft
The Netherlands

www.dcsc.tudelft.nl

Anonymous Referee #3
*Referee, Wind Energy Science*

Dear Referee,

First of all, the authors would like to thank the referee for his positive and constructive feedback. We believe that the comments have helped us to improve the quality of the paper. In our attempt to account for the comments, we have thoroughly revised different aspects of the paper. The objective of this document is to respond to the points raised by the referee and to provide a detailed overview of the changes made to the paper. The document consists of one section where we will respond to the review report provided by the referee.

Yours sincerely,

Sachin Navalkar
Mees van Vondelen
Alexandros Iliopoulos
Daan van der Hoek
Jan-Willem van Wingerden

Enclosure(s): Response to comments of Anonymous Referee #3

**Response to comments of Anonymous Referee #3**

- To motivate the problem statement in this publication, I suggest that the authors add a small paragraph indicating the sensitivity of turbines structural fatigue damage to damping. e.g. reducing structural damping by 5% results in xx reduction in turbine useful life time. You might need to do this for various components and materials, e.g. welded steel vs composites.
  The authors agree that such an explanation would emphasize the motivation. A figure indicating the total fatigue life of the turbine versus the damping has been included in the introduction.

- The authors discuss OMA for time-varying structures. However, OMA for non-linear and non-stationary time variant systems is not fully elaborated upon; e.g. use-case where such a dedicated treatment is needed: large deflections resulting in geometric non-linearity in long slender modern blades
  The authors agree that such an elaboration would enhance the explanation. A reference to a study as described by the referee and a small paragraph was included to shed light on this topic.

- I suggest that the authors further sub-group and elaborated on the OMA methods based on whether they require additional physics model representations and those that do not require it (e.g. model reduction techniques in OMA algorithms).
  Although the authors acknowledge the relevance of grey box methods, none of the investigated OMA algorithms in this study require additional physics model representations. The authors believe it is thus not an improvement to subgroup the methods in this study based on this principle. A short paragraph clarifying the focus on black-box methods and noting the existence of these grey-box methods is added.

- Controllers are crucial for modern wind turbines design. I suggest that the authors expand on the topic of OMA for system identification of structures with controller in the loop (closed loop identification).
  The authors agree with the statement. Next to the explanation about control damping already made in the paper, a paragraph was added to the introduction that explains that the application of OMA is valid for controlled systems such as operational OWTs, as that the ambient input is uncorrelated with the output. Furthermore, it was mentioned that the separation of the different damping sources is non-trivial and outside of the scope of this study.

- Page 8: OMA for structural modes above 5Hz and their corresponding damping might be in fact necessary when dealing with identifying a damaged structure. Expanding on this idea in this review would be a nice to have.

  The authors agree that OMA above 5 Hz might create opportunities for damage detection. However, this can most likely not be achieved in regular operating conditions, as suffucient strong excitation and sophisticated sensing equipment would be required to capture these modes. This explanation was added to the paper .